



# Differential absorption lidar measurements of water vapor by the High Altitude Lidar Observatory (HALO): Retrieval framework and validation

Brian J. Carroll[1], Amin R. Nehrir[2], Susan Kooi[3], James Collins[3], Rory A. Barton-Grimley[2], Anthony Notari[2], David B. Harper[2], Joseph Lee[2]

[1] NASA Postdoctoral Program Fellow, NASA Langley Research Center, Hampton, VA, USA
[2] NASA Langley Research Center, Hampton, VA, United States
[3] Science Systems and Applications, Inc., Hampton, VA, United States

*Correspondence to*: Brian J. Carroll, Amin R. Nehrir (brian.j.carroll@nasa.gov, amin.r.nehrir@nasa.gov)

**Abstract.** Airborne differential absorption lidar (DIAL) offers a uniquely capable solution to the problem of measuring water vapor (WV) with high precision, accuracy, and resolution throughout the troposphere and lower stratosphere. The High Altitude Lidar Observatory (HALO) airborne WV DIAL was recently developed at NASA Langley Research Center and was first deployed in 2019. It uses four wavelengths at 935 nm to achieve sensitivity over a wide dynamic range, and simultaneously employs 1064 nm backscatter and 532 nm high spectral resolution lidar (HSRL) measurements for aerosol and cloud profiling. A key component of the WV retrieval framework is flexibly trading resolution for precision to achieve optimal data sets for scientific objectives across scales. A technique for retrieving WV in the lowest few hundred meters of the atmosphere using the strong surface return signal is also presented.

The five maiden flights of the HALO WV DIAL spanned the tropics through midlatitudes with a wide range of atmospheric conditions, but opportunities for validation were sparse. Comparisons to dropsonde WV profiles were qualitatively in good agreement, though statistical analysis was impossible due to systematic error in the dropsonde measurements. Comparison of HALO to in situ WV measurements onboard the aircraft showed no substantial bias across three orders of magnitude, despite variance ($R^2 = 0.66$) that may be largely attributed to spatiotemporal variability. Precipitable water vapor measurements from the spaceborne sounders AIRS and IASI compared very well to HALO with $R^2 > 0.96$ over ocean and $R^2 = 0.86$ over land.

## 1 Introduction

Water vapor (WV) is a key component of the Earth's atmosphere and water cycle, playing major roles in cloud, weather, and climate processes, including radiative balance as the most dominant greenhouse gas (Trenberth et al. 2007). The need for accurate WV measurement across scales is widely recognized, as is the value of remote sensing for providing such measurements with desirable spatiotemporal coverage (Bony et al. 2006, Bony et al. 2015, Sherwood et al. 2010, Teixeira et al. 2021, Wulfmeyer et al. 2015).



Radiosonde networks have long provided the most consistent and extensive in situ WV profile measurements globally, though the operational network is limited by resources and personnel to a finite number of sites, only over land, and two launches per day (Ferreira et al. 2019). Spaceborne passive sounders have the great advantage of daily global coverage, and

their column and cloud products are important contributors to operational forecasting and climate research (Wulfmeyer et al. 2015). However, their WV retrieval vertical resolution is roughly $1-2$ km at best in the troposphere and accuracy is closely tied to the observed scene as well as the prior inputs to the retrievals. This makes spaceborne passive sounders incapable of capturing many lower tropospheric, cloud, and PBL processes relevant on the weather and climate timescales. Measurement techniques with higher spatiotemporal resolution, such as lidars, have become vital to studying WV processes as our

understanding improves and more demanding measurement criteria have become prevalent (Wulfmeyer et al. 2015). Due to the challenging nature of developing and deploying airborne lidars, there are not many within the research community. These sparse but capable airborne lidars are complimented with emerging technologies that show promise for enabling dense, low-cost surface-based networks (Nehrir et al. 2012, Spuler et al. 2015, Spuler et al. 2021). The mobility of airborne lidar cements it as an important asset for Earth science, able to observe large regions with high precision and resolution,

including over oceans where permanent observing networks are impractical.

Two types of lidar are commonly used for atmospheric WV measurements: Raman lidar (e.g., Cooney 1970, Eichinger et al 1999, Goldsmith et al. 1998, Leblanc et al. 2012, Philbrick 1994, Whiteman et al. 1992) and differential absorption lidar (DIAL; e.g., Browell et al. 1998, Ehret et al. 1993, Ferrare et al. 2004, Nehrir et al. 2011, Nehrir et al. 2017, Späth et al.

2016, Spuler et al. 2021, Wirth et al. 2009). Raman lidar systems provide the advantageous capability of monitoring multiple gas species simultaneously, but this comes at the cost of requiring large, high peak power ultraviolet lasers to overcome small Raman scattering cross-sections, and a need for frequent calibration. WV DIAL avoids the need for high peak power ultraviolet lasers, but only measures one species and requires the use of single frequency and frequency-agile pulsed lasers which have been the focus of decades of laser research within the DIAL community. A significant benefit of the DIAL

technique that results from the stringent laser transmitter requirements is the lack of a need for external calibration, as this technique relies on the ratio of elastic backscatter signals from a closely spaced wavelength pair, one online and one offline of a WV absorption line. The challenges of signal attenuation and limited dynamic range with a single DIAL wavelength pair can be overcome by utilizing multiple wavelength pairs spread along the side of the WV absorption line. The direct, calibration-free measurement of WV profiles with reduction in overall complexity resulting from emerging laser

technologies make WV DIAL a unique and important tool for atmospheric studies, and well-suited for airborne and space-based implementations.

Airborne WV DIAL systems were first employed in the 1980s (Browell 1983), with subsequent decades of literature documenting advances in the theoretical framework and instrument designs (e.g., Behrendt et al. 2007, Browell et al. 1998,



Ehret et al. 1993, Higdon et al. 1994, Ismail & Browell 1989, Wulfmeyer 1998) as well as measurements and applications (e.g., Carroll et al. 2021, Ismail et al. 2009, Kiemle et al. 2017, Schafler et al. 2021, Wakimoto et al. 2006, Wulfmeyer et al. 2006). Two airborne WV DIALs of particular note for this manuscript are the NASA Langley Research Center (LaRC) Laser Atmospheric Sensing Experiment (LASE, Moore et al. 1997) and the Deutsches Zentrumvfür Luft- und Raumfahrt (DLR) airborne WAter vapour Lidar Experiment in Space (WALES) instrument (Wirth et al. 2009). LASE is the

predecessor to the focus of this work, the High Altitude Lidar Observatory (HALO) WV DIAL. HALO has similar capabilities to WALES with simultaneous WV DIAL and high spectral resolution lidar (HSRL, Hair et al. 2008) capability for aerosols and clouds, however, HALO's modular design and exploitation of emerging laser and receiver technologies allows for a more compact form factor that permits operation on a wide range of aircraft.

This manuscript reports on the HALO WV DIAL retrieval framework and measurement capabilities based on its maiden flights in 2019, including sparse validation against the other instrumentation that was available. Section 2 describes the field campaign and instruments utilized in this work. Section 3 provides a brief review of WV DIAL principles followed by relevant specifics of HALO's design. Section 4 details the HALO WV retrieval methodology. Section 5 presents comparisons of HALO WV measurements with available in situ instrumentation and spaceborne sounders. Conclusions and

future directions are discussed in Section 6.

## 2 Instruments and comparison methodology

### 2.1 The NASA Aeolus Cal/Val Test Flight Campaign

The HALO measurements described in this manuscript were collected during the NASA Aeolus Cal/Val Test Flight campaign from 17-30 April 2019. The campaign was designed to provide calibration/validation comparisons with the

European Space Agency (ESA) space-borne Doppler wind lidar mission ADM-Aeolus (Stoffelen et al. 2005). The NASA Doppler Aerosol WiNd lidar (DAWN, Kavaya et al. 2014) provided wind profiles for this purpose, and HALO was deployed to validate aerosol measurements via HSRL as well as opportunistically test the WV DIAL capability. Further details of this campaign, including overviews of each flight, are given in Bedka et al. (2021). Data is archived and publicly available online (NASA/LARC/SD/ASDC, n.d.).


This campaign consisted of five flights of the NASA DC-8 aircraft: four from the NASA Armstrong Flight Research Center in Palmdale, California and one from Kona, Hawaii. These flights were mainly over the northeastern Pacific spanning midlatitudes to the tropics and observing a wide range of atmospheric conditions. There were only a few hours of observations over land. Installed along with HALO and DAWN on the DC-8 was the Diode Laser Hygrometer (DLH, Diskin

et al. 2002) to provide in situ WV measurements for HALO validation. Dropsondes were also deployed to validate wind and



WV measurements. This campaign provided the maiden flights for the HALO WV DIAL configuration which also employed simultaneous HSRL measurements of aerosols and clouds.

## 2.2 HALO

The HALO airborne lidar was developed at NASA Langley Research Center (LaRC) to address observational needs in the Earth Sciences, specifically focusing on atmospheric dynamics, composition, radiation, and the carbon cycle, in addition to serving as a technology testbed for future space-borne DIAL. It is a direct successor to the LASE airborne WV DIAL (Moore et al. 1997). HALO has a modular design with the capability to measure WV or methane mixing ratios via the DIAL technique along with aerosol, cloud, and ocean optical properties via the HSRL technique. This multi-functional design allows any two measurement capabilities simultaneously (i.e. WV DIAL/HSRL, WV DIAL/methane DIAL, or methane DIAL/HSRL) with rapid reconfiguration by using three modular laser transmitters and a single multi-channel, multi-wavelength receiver. Because of its compact design, the instrument can be flown on most research aircraft including autonomous operation aboard the high-altitude NASA ER-2. The methane DIAL/HSRL configuration has successfully flown in several field campaigns (e.g., Davis et al. 2021, Wu et al. 2021). This manuscript focuses on the WV DIAL retrieval framework and validation for its maiden flights during the Aeolus cal/val campaign. Details of the HALO instrument design and performance will follow in an instrument paper by Nehrir et al.

## 2.3 Dropsondes

The dropsondes deployed from the DC-8 were expendable digital dropsondes (XDD) with the Yankee Environmental Systems Inc. High-Definition Sounding System (HDSS). The HDSS and XDD are presented in full in Black et al. (2017) and will be referred to as sondes in this manuscript. The pressure, temperature, and humidity measurements were taken at 2 Hz, which is roughly 8 meters in vertical resolution. These sondes have been used in previous field campaigns (e.g. Doyle et al. 2017), but the Aeolus cal/val campaign was the first deployment with a new relative humidity sensor. This new sensor was found during the campaign to have a time-lagged response that appeared to vary with altitude, leading to large errors in the WV profiles (Bedka et al. 2021). This prevents a typical approach of using sonde measurements to quantitatively validate HALO WV profiles. Only a cursory qualitative comparison is made in Section 5. More rigorous comparison to dropsondes will be an objective of future HALO flights.

## 2.4 DLH

The NASA Langley/Ames DLH (hereafter just DLH) is designed to measure in situ WV in the troposphere and lower stratosphere while flying on board research aircraft, in this case the NASA DC-8 (Diskin et al. 2002). It reports WV mixing ratio at 1 Hz and is considered to be a community standard for accurate measurements of WV from the upper troposphere and lower stratosphere (UT/LS) down to the surface. The measurement relies on WV absorption of diode laser light in the 1.4 µm spectral region, tuning to either a weak or strong absorption line to accommodate the wide range of WV





concentrations in the atmosphere. It uses an open-path double-pass configuration between the laser transceiver inside the DC-8 pointing out a modified window panel and a retroreflecting panel mounted on an outboard engine nacelle. The calibration, algorithm, and validation study by Podolske et al. (2003) found a 1σ error estimate of 3.7%. This qualifies the

DLH as an excellent validation instrument for the HALO WV measurements, although conclusions are limited by the instruments' different sampling volumes.

Comparisons between HALO and the DLH were made in two ways for this study. The first was a single 10 km spiral descent DLH profile with preceding overpasses for remote sensing profiles by HALO. The second was considering the closest

HALO measurements temporally and vertically to the DLH flight altitude measurements (i.e. the highest-altitude HALO measurement) for all available data from the campaign. The closest comparable HALO measurement was typically ~400 m below the aircraft due to geometric overlap of the transmitter and receiver. Over this vertical distance, some atmospheric variation can be expected in addition to a systematic bias of higher WV mixing ratios with decreasing altitude, but these effects should be small on average given the predominant 8 – 12 km flight altitude during the campaign. This comparison is

a good opportunity to validate HALO's measurements of upper tropospheric WV, which is a region rarely profiled with the accuracy, precision, and resolution of a lidar.

## 2.5 Satellites

The Atmospheric Infrared Sounder (AIRS) and Infrared Atmospheric Sounding Interferometer (IASI) are considered community standards for spaceborne WV measurement and are used here for comparison to HALO WV products. The AIRS

and IASI precipitable water vapor (PWV) products were validated against ground-based measurements by Roman et al. (2016), finding that both satellites generally fell with a 5% error range, except for very dry (< 5 mm) or very moist (> 50 mm) regions which tended to have larger wet or dry biases, respectively.

### 2.5.1 Atmospheric Infrared Sounder (AIRS)

The AIRS instrument onboard the polar-orbiting NASA Aqua satellite retrieves temperature, humidity, and trace gas

information with daily global coverage to support weather prediction, study the water and energy cycles, and provide a record of several critical greenhouse gases (Chahine et al. 2006, Le Marshall et al. 2006). The hyperspectral sounder measures infrared radiation from 3.7 to 15 µm in a cross-track scanning pattern with a 13.5 km diameter nadir field of view (FOV), stretching to 22.4 km along-track by 41.0 km cross-track at scan edge. AIRS overpasses were suitable for comparison to HALO on 25 and 27 April, both passing through the ITCZ.


The AIRS Level 2 Version 7 data products used in this work are derived from 3-by-3 arrays of AIRS FOVs, and uncertainty estimates are provided with each retrieval (Thrastarson et al 2020). Column PWV and mixing ratio profiles were compared to HALO. The temporal average and standard deviation of HALO measurements within 25 km of the AIRS footprint center



were used to mitigate the instrumental and spatiotemporal sampling differences between the instruments, similar to previous

studies (e.g., Bedka et al. 2010, Diao et al. 2013). HALO records were omitted from PWV comparison if a cloud was

detected or the aircraft was below 8 km. The AIRS profiles used here are from the support data files which have 100 vertical

levels, but it is important to note that the 100 levels represent a much finer resolution than the actual independent

information within the profile. AIRS WV vertical resolution is actually limited to 1-2 km at best within the troposphere

(Gettelman et al. 2004, Thrastarson et al 2020, Wong et al. 2015, Wulfmeyer et al. 2015).

### 2.5.2 Infrared Atmospheric Sounding Interferometer (IASI)

The IASI instrument onboard the polar-orbiting ESA MetOp satellite series is designed to support numerical weather

prediction (NWP) by providing information on temperature, humidity, and some trace gases with global coverage twice per

day (e.g. Clerbaux et al. 2009, Hilton et al. 2009, Hilton et al. 2012, Klaes et al. 2007). IASI is composed of an imaging

system and a Fourier transform spectrometer to analyze infrared spectra between 3.6 µm to 15.5 µm. IASI scans across-track

with 30 elementary fields of view (EFOV), each of which contain a 2-by-2 grid of 4 instantaneous fields of view (IFOV).

The IFOV footprint is a circle of 12 km diameter at nadir and an ellipse of 20 km along-track by 39 km across-track at swath

edge. While IASI humidity is reported at up to 101 vertical levels, there are only a maximum of 10 independent pieces of

information within the profile and sensitivity is lowest in the lowest few kilometers of the troposphere, which is also the area

of highest moisture content.


The IASI data used in this work is the operational Level 2 data Version 6 for the instrument onboard MetOp-B. The data

files report PWV and WV mixing ratio profiles for each IFOV, but do not provide associated uncertainties beyond the

instrument accuracy requirement of 10%. To provide some estimate of uncertainty related to spatiotemporal variability in

comparison to HALO, the mean and standard deviation of the 4 IFOV products within a given EFOV were calculated and

used here. Likewise, HALO measurements within a 20 km radius of the EFOV center are averaged for the comparisons. This

approach is similar to past studies (e.g., Chazette et al. 2014, Roman et al. 2016). Overpasses were suitable for comparison to

HALO on 17, 22, 25, and 27 April, which included the tropics and midlatitudes.

## 3 DIAL principle and HALO measurements

### 3.1 DIAL theory

Derived from the lidar equation, the DIAL equation can be written as

$$N\left(r + \frac{\Delta r}{2}\right) = \frac{1}{2\,\Delta r\,\Delta\sigma(\lambda_{on},\lambda_{off},r)}\ln\left(\frac{P(\lambda_{on},r)}{P(\lambda_{on},r+\Delta r)}\frac{P(\lambda_{off},r+\Delta r)}{P(\lambda_{off},r)}\right), \tag{1}$$

where $N$ is the number density of WV (molecules/cm$^3$), $r$ is range from the lidar (cm), $\Delta r$ is the range cell length over which

$N$ is calculated, $\Delta\sigma$ is the difference between online and offline absorption cross sections (cm$^2$), $\lambda$ is the online or offline





wavelength as indicated by subscript, and $P$ is the measured backscatter signal (Schotland 1974). This dependence on the

ratio of online to offline signals yields a direct, calibration-free measurement of WV assuming the following conditions are true: the spectral separation between the on and off wavelengths in Eq.(1) is small enough to neglect differences in atmospheric scattering and transmission properties (except absorption by WV), the spatiotemporal variations in the atmosphere are negligible between the two laser pulses, the pulsed laser exhibits high spectral purity (Ismail & Browell 1989), Doppler broadening is constant across $\Delta r$, and instrument differential transmission is constant within a retrieval

(Schotland 1974). It is also assumed that the WV concentration in the retrieval bin is constant.

In practice, accurate measurements require careful consideration of $\Delta\sigma$ calculation, corrections to the measured signals (e.g., Doppler broadening of backscattered light), and close monitoring of instrument and laser characteristics. Speaking more generally, potential sources of error in the DIAL measurement are statistical, systematic, or stem from uncertainties in $\sigma(\lambda,r)$.

These sources of error have been presented at length in the literature (Schotland 1974, Remsberg & Gordley 1978, Ismail & Browell 1989, Wulfmeyer & Bosenberg 1998) and reproduction is unwarranted here. Some details of implementation with HALO will be discussed further in Section 4 and a future instrument paper by Nehrir et al. We will highlight here the sensitivity of statistical uncertainty to the temporal (or along-track) and range resolutions of the DIAL measurement:

$$\delta N \propto (\Delta t)^{-0.5}(\Delta r)^{-1.5}, \tag{2}$$

where $\delta N$ is statistical uncertainty in the DIAL measurement of $N$ and $\Delta t$ is temporal resolution (Ismail & Browell 1989). This proportionality indicates that the statistical uncertainty of a DIAL measurement can be reduced by decreasing resolution, with greater sensitivity to changes in $\Delta r$ than $\Delta t$. This is caused by Poisson statistics in both dimensions giving the 0.5 exponent along with the differential absorption along the path length of $\Delta r$. Equation 2 assumes validity of Poisson sampling statistics for the DIAL system in question, which is typically applicable to photon counting systems but can also be

applied to analog detection systems such as HALO via a noise scale factor (NSF, Liu et al. 2006).

Optical depth (OD) is fundamental to the DIAL principle, affecting signal-to-noise ratio and factoring into the $\delta N$ calculation (Wulfmeyer & Bosenberg 1998). The OD between the instrument and range $r$ can be calculated in a very straightforward manner from the DIAL signals:

$$OD = \frac{1}{2}\ln\frac{P(\lambda_{off},r)}{P(\lambda_{on},r)}, \tag{3}$$

assuming as in Eq. 1 that online and offline wavelengths are spectrally close enough to have identical transmission except for the absorption due to WV. The optimal one-way OD at the online wavelength for an ideal DIAL system is 1.1, after which the signal becomes too attenuated for accurate measurement (Remsberg and Gordley 1978). However, the exact value for a realized system is dependent on many system parameters such as the detection noise floor and pulse energy, and thus in

practice the maximum one-way WV OD is typically in the range $1 - 2$, and for a single range cell is $0.01 - 0.05$ (Bosenberg





1998, Wulfmeyer & Bosenberg 1998). To meet these criteria, flexibility in selection of online wavelengths and $\Delta r$ are incorporated into the HALO WV retrieval.

**3.2 HALO water vapor DIAL**

HALO transmits four wavelengths around 935 nm to cover the large WV dynamic range from the moist surface to the dry

UT/LS. These four wavelengths are spread between varying strength absorption features and wings of lines in the 935.5 nm line complex as shown in Fig. 1a, thus providing sensitivity across the wide dynamic range of WV within the troposphere and UT/LS. We refer to the most strongly absorbed HALO wavelength as $\lambda_1$, progressing sequentially to $\lambda_4$ as the least-absorbed wavelength. The strong absorption of $\lambda_1$ makes it an ideal online wavelength in very dry UT/LS, suitable for ~8-20 km altitude range (depending on aircraft altitude) and therefore compatible with high-altitude aircraft such as the ER-2. $\lambda_2$

follows as the typical online wavelength for the mid-troposphere, and $\lambda_3$ is chosen for the lower troposphere and PBL. Due to the high variability of lower tropospheric moisture (e.g., tropics versus arctic), $\lambda_3$ is tunable within a 22 GHz range (0.064 nm) when referenced to $\lambda_2$. $\lambda_4$ is offset locked to $\lambda_3$ and thus tunes in tandem. The details of this tunability and transmitter design will be explored in a subsequent instrument paper.

In selecting online and offline pairs for a given DIAL calculation, consecutive wavelengths are used, e.g., $\lambda_2$ online and $\lambda_3$ offline. This optimizes accuracy concerning the DIAL equation assumptions by minimizing spectral differences in transmission and scattering along with spatiotemporal differences in the sampled volume between shots since the $\lambda$ switching is sequential. Figure 1b shows an example nadir profile of lidar signals at the four wavelengths, attenuating with increasing range from the aircraft. Each wavelength pair is sensitive to the altitude range that has sufficiently high differential

absorption optical depth (DAOD) and sufficient online signal strength, which is demonstrated with the approximate random error plots in Fig. 1c. The increase in signal below 1 km in Fig. 1b is due to enhanced aerosol backscatter within the PBL. Figure 1b also shows the low end of the dynamic range of the detectors, with $\lambda_1$ and $\lambda_2$ reaching the noise floor around $10^0$ counts. It should be noted that the plots of Fig. 1b and 1c will differ from profile to profile based on the vertical distribution and magnitude of the WV profile as well as the aerosol loading and scattering properties within the sampled volume. The

precision of the WV DIAL retrieval is also highly dependent on instrument parameters and spatial averaging.

The HALO WV DIAL uses an injection seeded and frequency doubled Nd:YAG laser to pump an injection seeded optical parametric oscillator (OPO) to generate output at 935 nm with 1 KHz pulse repetition frequency (PRF). The 935 nm seed laser and OPO cavity are tuned to switch between the four wavelengths on a shot-by-shot basis. The residual 532 nm and

1064 nm pulsed energy left over from the OPO conversion process is transmitted collinearly with the 935 nm output to enable simultaneous aerosol/cloud profiling utilizing HSRL (532 nm) and backscatter (1064 nm) techniques, including depolarization measurement at 532 and 1064 nm. The simultaneous WV and HSRL sampling approach employed with HALO is similar to previous work (e.g., Wirth et al. 2009) but the design to utilize a single transmitter for all wavelengths

makes HALO a uniquely capable and compact instrument with the ability to support airborne campaigns from a wide range

of aircraft optimized for different sampling strategies.

Real-time onboard processing of the 1 KHz PRF signal sums shots at each wavelength to improve the signal-to-noise ratio (SNR) and reduce the data rate to 2 Hz. The data system sampling rate is 120 MHz or 1.25 m in range, but the WV DIAL channels are limited by the detection chain electrical bandwidth to 15 m vertical resolution. The high vertical resolution

allows for oversampling of surface or cloud signals to allow for high spatial resolution total and partial WV columns. To keep the file sizes manageable, the atmospheric signals are filtered and downsampled to 15 m vertical resolution. Subsequent temporal and vertical averaging of the WV DIAL data are employed to improve the precision of the DIAL retrieval, and this is discussed in further detail in Section 4. The HSRL signals have sufficient bandwidth to maintain 1.25 m vertical resolution, which will be used for future cloud and ocean profiling measurements, but the atmospheric data shown in this

manuscript are digitally filtered and binned to 15 m vertical resolution in post-processing to increase SNR and match the WV DIAL data resolution.

The receiver dynamic range is extended by implementing a high and low optical split. Dynamic range is further increased as both optical channels have an adjustable dual-gain (high and low) electronic output. The high-optical high-electrical gain

channel is used for all atmospheric measurements in this manuscript and will be referred to simply as the "high gain" channel. The low-optical high-electrical gain channel is used in surface return calculations and will be referred to as the "low gain" channel. The other channels are used for diagnostics and will be optimized for measurements in future campaigns.

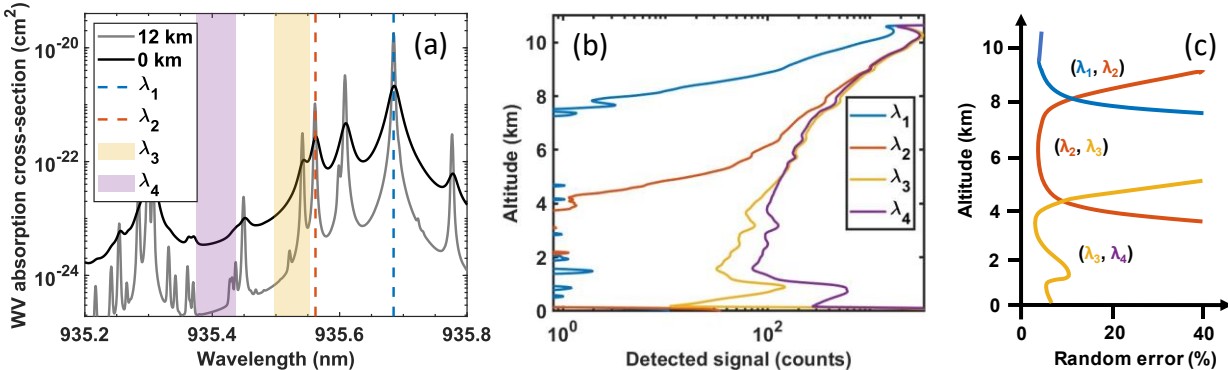

**Figure 1. (a) Voigt spectrum of WV absorption cross-section at 12 km and at sea level, with vertical lines and shading indicating the fixed and tunable HALO wavelengths. (b) Example profile of HALO measured signals for the four transmitted wavelengths, averaged temporally and vertically to match a typical final WV product resolution (60 seconds temporal, 315 m vertical). (c) Resultant sensitivity of line pairs to different parts of the atmosphere, expressed as an estimation of random error versus altitude. Line pairs are in parentheses as (online, offline).**






Some key parameters of the HALO instrument and processing are listed in Table 1 and will be discussed in greater detail below. Another relatively brief summary of HALO WV DIAL was presented in the Aeolus cal/val campaign overview by Bedka et al. (2021), and further technical details will be presented in an instrument paper by Nehrir et al.

**Table 1. HALO parameters.**

| Transmitter type | Custom Fibertek Nd:YAG pumped OPO |
|---|---|
| Wavelengths | 532, 935, 1064 nm |
| Pulse energy: 532, 935, 1064 nm | 6, 1.5, 6 mJ |
| PRF | 1000 Hz (effectively 250 Hz for each DIAL λ) |
| Average power: 532, 935, 1064 nm | 6, 1.5, 6 W |
| Measurement principal: 532, 935, 1064 nm | HSRL, DIAL, backscatter |
| Detector type: 532, 935, 1064 nm | PMT, APD, APD |
| Telescope Diameter | 40 cm |
| Receiver FOV (DIAL, HSRL) | 300, 1000 μrad |
| DIAL temporal resolution * | 5 – 60 s ** |
| DIAL vertical resolution * | 250 – 585 m |
| DIAL measurement dynamic range | 0.001 – 25 g/kg |

*Resolution is variable, with increasing statistical uncertainty accompanying finer resolutions, as discussed in Section 4A.
**For typical NASA DC-8 flight speed, including the data this paper, this is roughly 1 – 12 km.

## 4 HALO water vapor retrieval methodology

The HALO DIAL data processing and WV calculation is based on the preceding decades of DIAL research within the
community and implements heritage techniques developed for HALO's predecessor LASE (e.g., Ismail & Browell 1989, Moore et al. 1997). This section gives an overview of the HALO WV retrieval, with additional details in subsections where warranted.

Two components of the DIAL equation (Eq. 1) require extensive consideration: the lidar signals and the absorption cross-
sections. Their treatment is outlined in Fig. 2. First, electronic and atmospheric background signals are removed from the raw lidar signals before digitally filtering and downsampling to 15 m range resolution. Data below cloud top or terrain are then removed based on cloud-top heights (CTHs) identified from HSRL and terrain elevation from the Global Land One-Kilometer Base Elevation (GLOBE, Hastings & Dunbar 1999) digital elevation model, respectively. An additional 45 m back-off is added to both cloud and terrain to account for any spatial heterogeneity within the observation time. High-altitude





cirrus clouds are not automatically masked because they are often thin enough to be penetrated. Manual inspection of final
datasets is employed for any additional masking that may be necessary.

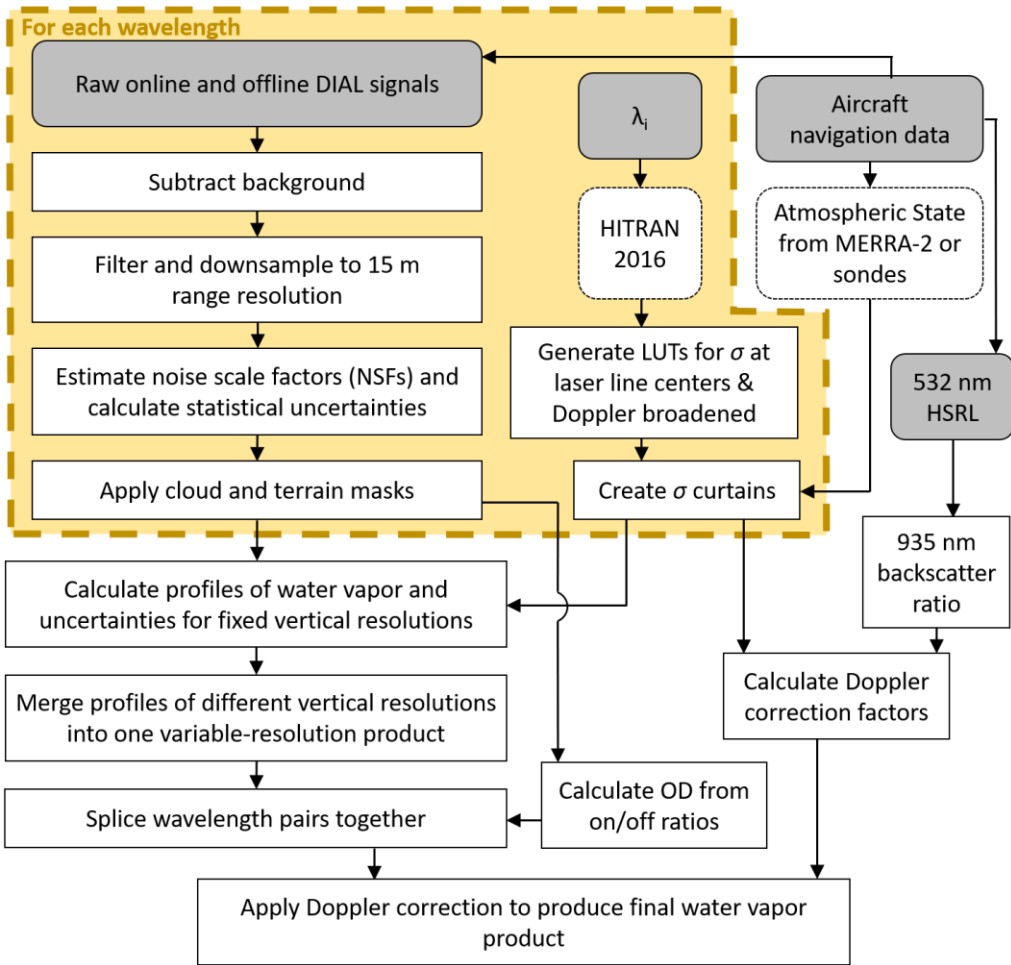

**Figure 2. Flowchart of the HALO WV retrieval steps. Gray boxes are data or user inputs.**


Calculation of absorption cross-section profiles with proper consideration of collisional (pressure) and Doppler broadening is
critical for accurate DIAL measurements. This begins with the creation of lookup tables (LUTs) of WV absorption cross-
section for each transmitted HALO wavelength over the reasonable expected range of pressures and temperatures. These
LUTs utilize the 2016 High Resolution Transmission database (HITRAN, Gordon et al. 2017). The atmospheric state for

each profile is taken from the MERRA-2 reanalysis model (Gelaro et al. 2017), or radiosondes/dropsondes when MERRA-2
is not yet available during field campaign operation. The atmospheric state at each point gives the pressure and temperature
for the LUT, and thus the absorption cross-section curtains are generated for the DIAL calculation. One consideration not



shown in Fig. 2 is an additional step to address collisional self-broadening, wherein WV absorption line broadening is a function of the WV mixing ratio. The contribution of this self-broadening is first calculated with WV mixing ratios from
MERRA-2, then the resultant HALO WV mixing ratio product is fed back into the absorption cross-section curtain creation to fine-tune the contribution of self-broadening. The subsequent code is then rerun to produce the final HALO WV product.

Once the lidar signals and $\Delta\sigma$ have been implemented in Eq. 1 to calculate WV number density, the collocated MERRA-2 molecular number density is used to convert the HALO backscattered power measurements to WV mass mixing ratio. This
is done for each of the three wavelength pairs and at multiple range resolutions, and those resolutions are then combined into a single curtain of WV measurements for each wavelength pair (Section 4A). Wavelength pairs are then spliced together for each profile based on optical depths (Section 4B), and surface return measurements are appended below the minimum altitude of atmospheric calculations (Section 4C). Lastly, Doppler correction factors are applied to produce the final WV curtain (Section 4D).


Parallel to the lidar signal processing is the calculation of statistical uncertainty in the WV product. This uncertainty is the statistical error based on Poisson statistics (e.g., Eq. 9 of Ismail & Browell 1989) adapted to the analog detection used by HALO via NSFs (Liu et al. 2006). The uncertainties calculated for each wavelength pair and range resolution are merged into one final curtain in the same manner as the WV mixing ratios.

**4.1 Variable resolution**

As discussed in Section 3A, uncertainty in a DIAL measurement can be reduced by using a coarser temporal (horizontal) or range (vertical) resolution. This allows for flexibility in processing to optimize for a given scientific objective. HALO WV processing is typically done at a fixed temporal resolution with vertical resolution that varies between two values, because the statistical uncertainty of the DIAL measurement is more sensitive to the vertical resolution. For most of the HALO data
shown in this manuscript, the initial DIAL calculation vertical resolution is 315 m. If the statistical uncertainty for a given mixing ratio value at 315 m resolution exceeds 6%, the value is replaced with the corresponding 585 m vertical resolution value. These vertical resolutions are user-defined and can be made finer or coarser when reprocessing the data, as can the conservative choice of 6% uncertainty threshold. 6% was chosen for this dataset by empirical investigation to provide precise measurements even in the most challenging environments. A linear weighting function is applied to a 165 m vertical
window centered on each bin where the range resolution changes to ensure a smooth transition in the WV profile.

To illustrate how the chosen resolutions for the DIAL calculation enable applicability to science targets across scales, Fig. 3 explores the WV product and its statistical uncertainty for temporal resolutions ranging from 1 second to 70 seconds with fixed vertical resolution, either 315 m or 585 m. This temporal range corresponds to 200 m to 14 km along-track for the
typical DC-8 flight speed in the data shown here but would differ when HALO is deployed on other aircraft. Figure 3c and





3d exemplify Eq. 2, that statistical uncertainty for a given measurement decreases proportional to $\Delta t^{0.5}$ and that the statistical uncertainty is more sensitive to $\Delta r$ than $\Delta t$ because a larger $\Delta r$ results in a larger DAOD and the precision of the WV retrieval is directly proportional to the DOAD. It should be noted that Eq. 2 holds for any given profile, but the appearance of plots such as those in Fig. 3 will vary depending on the scene. For example, this profile shows a jump in uncertainty

below ~4.5 km, where the $\lambda_3$ / $\lambda_4$ pair is used, because $\lambda_3$ for this flight was optimized for wetter environments than were observed at that time. Since this paper is not investigating specific targets, a conservative choice of 60 s temporal resolution has been implemented in the retrieval, except where stated otherwise.

**Figure 3. Plots showing (a, b) WV mixing ratio and (c, d) the associated statistical uncertainty for different temporal and range resolutions for a single profile centered at 25.04 UTC from the 2019-04-29 flight (this profile is also shown in Fig. 4). (a, c) have $\Delta r$ = 315 m throughout the profile while (b, d) have $\Delta r$ = 585 m. The x-axis indicates different temporal resolutions ($\Delta t$), which translate to along-track horizontal resolution calculated for the typical DC-8 flight speed.**





Figure 4 shows HALO data for a section of a midlatitude flight. Figure 4a is the final WV mixing ratio product, with corresponding statistical uncertainty and range resolution shown in Fig. 4b and 4c. Various environmental conditions and features were sampled, including broken and unbroken marine stratocumulus, a very dry layer above the PBL, and moist layers throughout the troposphere that in one location extended towards the tropopause (visible and infrared satellite imagery for this flight was shown in Bedka et al. 2021). In Fig. 4b, many of the sharper gradients in uncertainty are a result

of a change in vertical resolution (compare to Fig. 4c). The largest uncertainties for this scene were driven by insufficient DAOD in the dry layer above cloud-top. Because the layer was very dry, the small DAOD was difficult to measure with the weakly absorbed $\lambda_3$. Aerosol scattering in the PBL provided good SNR and thus low uncertainties. Aircraft altitude also indirectly affected uncertainty by effectively shifting the SNR profile; this can be seen around 0.4 – 0.9 UTC. Figure 4d exhibits the utility of the simultaneous HSRL measurements to identify clouds and the PBL while providing valuable

information for WV-aerosol-cloud interaction studies.

## 4.2 Splicing profiles from multiple wavelength pairs

After the variable range resolution has been determined for each wavelength pair, the WV profiles of the three wavelength pairs are spliced together based on WV DAOD thresholds. The extent of each splicing region is plotted in Fig. 4c as an example. Starting from the highest altitude, WV calculated from the first wavelength pair ($\lambda_1$ and $\lambda_2$) is used until a DAOD of

1.0 is reached, at which point a linearly weighted average incorporates an increasing contribution from the second wavelength pair until an OD of 1.6. The second wavelength pair is then used alone until the OD range 1.0 to 1.5, wherein again a linearly weighted average controls the transition to the third wavelength pair. The third wavelength pair is used alone for the rest of the profile. The OD thresholds can change from these default values based on manual inspection of instrument performance and atmospheric scene.




**Figure 4.** Time-altitude curtains of HALO (a) WV mixing ratio, (b) percent uncertainty, (c) vertical resolution, (d) HSRL 532 nm aerosol backscatter on 30 April. DLH measurements are also shown in (a) at the aircraft altitude, whereas aircraft altitude is a magenta line in the other panels. The white lines in (c) mark wavelength pair and splicing regions, as indicated.




### 4.3 Near-surface water vapor measurement via surface return signals

The center of a nadir-pointing atmospheric DIAL measurement window (i.e., the altitude at which the value is reported) is by its nature limited to a distance of $\Delta r$, the range resolution, above a hard target scattering surface (e.g., land, ocean, or cloud top). As discussed above, $\Delta r$ is driven by the measurement SNR and desired precision. However, the unresolved lowest bin
(from the surface up to an altitude of $\Delta r$) can be retrieved by carrying out a DIAL retrieval using the strong surface return and the last atmospheric bin above the surface. This is accomplished by employing Eq. 1 with $\lambda_3$ and $\lambda_4$ where $P(\lambda, r + \Delta r)$ is replaced with $P(\lambda, r_{surface})$.

These signals are depicted in Fig. 5, utilizing high gain and low gain signals for $P(\lambda, r)$ and $P(\lambda, r_{surface})$, respectively. The
high gain atmospheric signals are temporally and vertically averaged matching the typical atmospheric DIAL resolutions of 60 s and 315 m (i.e., the colored high gain points in Fig. 5). Due to the increase in SNR from aerosol scattering in the PBL, future work may reduce this averaging to a finer resolution. Only atmospheric signals more than about 100 m above the surface are considered to ensure no contamination from surface heterogeneity or other effects such as sea spray. The low gain channels have an optimal receiver dynamic range for the surface return signals, ensuring that the strong signal is
captured within the linear regime of the channel's digitizer. To ensure accurate representation of the energy capture within the impulse response from the surface return, the data from 5 bins centered about the peak of the surface return are integrated (highlighted with color in Fig. 5). The number of integrated bins was empirically determined and little improvement was observed as the number of integrated bins was increased. The online and offline surface return signals were ratioed then temporally smoothed to 60s to match the atmospheric data. This order of operations was empirically determined as the best
approach for reducing outliers.

While this method was previously employed with LASE and is similar to other techniques (e.g., integrated path differential absorption (IPDA) lidar (Barton-Grimley et al. 2021)), to the authors' knowledge it has not been previously published with application to range-resolved WV DIAL profiling. This method has only been applied to HALO for clear-sky data over
oceans thus far because it is a relatively uniform surface compared to cloud or land. Moving forward, the surface return methodology will be adapted to extend HALO measurements down to those more complex surfaces. Topographic and albedo variability of cloud and land will require a more detailed treatment to ensure surface-related changes in signal are separated from atmospheric OD variation. The HALO low gain channels will be optimized to keep the signals over clouds and land on scale, something that was not the focus of the maiden HALO flights. Furthermore, the surface return will be
examined with the native HALO 1.25 m resolution, which despite being oversampled still allows for much more accurate peak finding and representation of the energy captured within the surface impulse response. This latter topic is an area of ongoing research and will be the subject of a follow-on study.



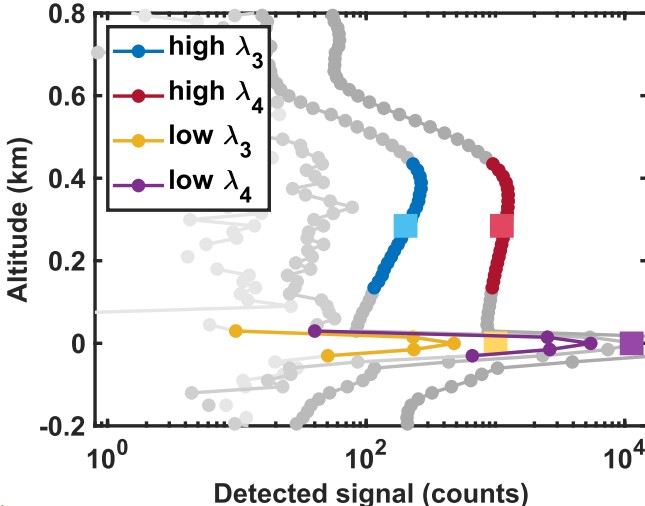

**Figure 5. Illustration of surface return DIAL concepts. The legend "high" and "low" refer to high gain and low gain channels,**
**plotted at 15 m vertical resolution. The high gain data is temporally averaged to 60 sec, while low gain is a single half second averaged record. The colored circles are averaged (atmospheric) or summed (surface return) to use in the DIAL calculation. These are the $P(\lambda, r)$ inputs to Eq. 1 and are shown as colored squares on the plot. Grayscale points are not used in the surface return but are still shown here for context.**

## 4.4 Doppler broadening correction


Another consideration in optimizing the HALO WV retrieval is accounting for the Doppler broadening of the backscattered signal. The backscattered light from aerosols experiences negligibly small Doppler shifts (MHz), whereas the backscattered light from molecules is significantly Doppler-broadened (GHz) because of the temperature-dependent molecular velocity distribution. The Doppler-broadened return signal must be considered carefully due to the wavelength-dependent OD of the

return path. In particular, the signal from the start of a range cell and the end of that range cell can experience different aerosol ODs based on the aerosol gradient within that $\Delta r$. Thus, the aerosol backscatter ratio profile is necessary for correct consideration of Doppler broadening effects, and in this regard the HALO HSRL measurements are a unique advantage of the HALO architecture. A full description and mathematical framework of the Doppler broadening correction for WV DIAL measurements is presented in Ansmann (1985) and Ansmann and Bosenberg (1987). The correction's application to HALO

was streamlined by experience with HALO's predecessor LASE (Ismail & Browell 1989).

Doppler broadening correction is implemented in the HALO retrievals beginning with creation of 2 LUTs for absorption cross sections with the transmitted wavelength, pressure, and temperature as user inputs to the Voigt function which utilizes the 2016 HITRAN database for fundamental line parameters. The first LUT is the same as the single-wavelength lookup

tables described above and is used for the aerosol portion of the backscatter. A second LUT is comprised of the Doppler (Gaussian) weighted sum of the Voigt calculated absorption cross sections across a range of +/- 0.05 wavenumbers from





each transmitted laser wavelength. From these 2 LUTs, 2 curtains of absorption cross sections for each transmitted wavelength are produced, one for aerosol scattering and one for molecular scattering. The HALO 532 nm aerosol backscatter ratio derived using the HSRL technique (Hair et al. 2008) is extrapolated to the DIAL wavelengths using the wavelength

dependence of the molecular scattering and the backscatter angstrom exponent to estimate the 935 nm aerosol backscatter ratio (Burton et al. 2012). This ratio is used to determine a linearly weighted combination of the Mie and Rayleigh absorption cross section curtains in the Doppler broadening calculation. The DAOD for the given DIAL wavelength pair is calculated using the combined curtain to determine a Doppler correction factor. The correction factor is applied to the WV mixing ratios that were previously retrieved without consideration of Doppler broadening to yield the final WV product.

**5 Data product comparisons and discussion**

Sondes were deployed during the Aeolus cal/val campaign to validate the wind measurements by Aeolus and DAWN. The sondes also opportunistically provided WV measurements for comparison to HALO WV DIAL. However, due to the poor performance of the sonde WV measurements, only a very limited qualitative comparison is possible (discussed further in Bedka et al. 2021). This section therefore places emphasis on HALO WV comparisons to the other observations that were

available, namely the DLH and satellites, which proved useful despite lacking the profiling spatiotemporal resolution to constitute an ideal validation dataset.

**5.1 Dropsondes**

A representative set of WV profiles from sondes and HALO are shown in Fig. 6, taken throughout the campaign including 4 out of the 5 flights, spanning the tropics to relatively dry midlatitudes. HALO profiles with 30 s and 60 s temporal resolution

are shown to further exemplify the resolution flexibility discussed above. Vertical resolution is the same for both, using the default algorithm for variable vertical resolution applied to the 60 s dataset. Both resolutions have very good agreement with the sondes and with each other, as expected from Fig. 3. As mentioned in Section 2C only a qualitative comparison to dropsondes is possible from this campaign because the dropsonde moisture sensors had a nonlinear damped response and erroneously low values in the first few km of descent. This damped response is seen as a vertical shift, i.e., for a given

feature in the HALO data, the same feature is in the sonde data but its location is shifted downwards by a variable amount (this is especially clear in Fig. 6b, 6c). This shift starts large then diminishes in an inconsistent way, making a correction infeasible and limiting to qualitative comparison.





**Figure 6.** HALO WV mixing ratio profiles with statistical uncertainty bars, and collocated dropsonde profiles for comparison. The bars have been decimated for legibility. The sonde WV sensors had erroneous nonlinear damped responses and thus cannot be used for a rigorous quantitative comparison.





In the middle and lower troposphere, the dropsondes and HALO clearly resolved the same moisture structures with very similar results across two orders of magnitude, approximately 0.1 to 20 g/kg. Dry layers within the lower troposphere were captured by both instruments. The lowest few hundred meters where the HALO measurements utilized the surface return

signal also showed good agreement, with differences typically less than 15%. This supports the validity of the surface return WV retrieval. Across the wide range of aircraft altitudes and environmental conditions that were sampled, there were no apparent systematic biases or other deficiencies in HALO WV for the atmospheric signal profiles or the surface return retrieval.

## 5.2 DLH

The DLH onboard the DC-8 was the best available option for validation of a HALO WV profile. Only one spiral descent was carried out during the campaign, from 1.34 UTC to 1.82 UTC on 30 April (shown in Fig. 4). This DLH profile from 10 km down to 160 meters above the surface, with an average descent rate of ~1300 ft/min, is shown alongside HALO profiles in Fig. 7. DLH data is shown at native resolution and with 315 m vertical smoothing to match the HALO measurements. The DC-8 descent occupied 50 km$^2$ and the preceding overpasses were tangential to this area due to vectoring by air traffic

control, so spatiotemporal differences in the measured profiles are inevitable and there is no single best HALO profile to compare to the DLH. Furthermore, the DC-8 in-situ spiral diameter was approximately a quarter of the length of the remote sensing overpass leg and substantial variability was observed by HALO within this volume, which could explain the high frequency variability in the DLH data around 4 km. The gray shaded area in Fig. 7 shows the range of values measured by HALO in the overpass preceding the spiral, from 0.93 to 1.06 UTC, which included 8 independent profiles. An example

HALO profile taken near the start of the descent is also shown (1.51 UTC), with data above 7 km appended from a slightly earlier profile with a higher aircraft altitude (1.38 UTC). The vast majority of the DLH profile was within the range of values observed by HALO, with two very small discrepancies in the upper troposphere and another at PBL top, which were potentially caused by spatiotemporal variability between the measurements. The single HALO profile also shows good agreement, exemplifying the measurement precision across the wide dynamic range of the profile and even capturing the

very dry layer above the PBL, which is particularly challenging from a range of several km.

A sonde profile is also shown in Fig. 7. The sonde was launched at 1.43 UTC, near the start of the spiral descent. The sonde is not in agreement with the DLH nor the range of HALO values until the lower half of the profile, and even then appears to have temporal lag as it exceeds the range of HALO values around the sharp gradients below 3 km. This reinforces the

previous statements that the sondes were not reliable enough for quantitative validation of HALO WV.

We do not present error statistics for this one DLH profile because the dataset is too small, and differences are likely dominated by specific atmospheric features and sampling differences between the instruments. Many profiles would be needed (e.g., from DLH or functional sondes) to objectively assess precision or any systematic bias in the HALO profiles.




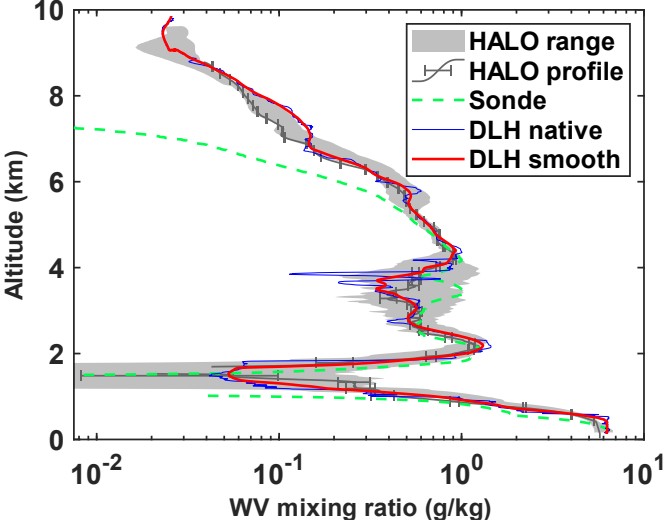

**Figure 7. HALO comparisons to DLH and sonde from the descent on 04/30. The blue line is the DLH data at native resolution. The red line is DLH smoothed with a 315 m rolling average to match the constant 315 m vertical resolution of the HALO profile. The gray area shows the range of HALO measurements in the area of the spiral descent from the preceding overpass, indicating the expected spatiotemporal variability of the region. An example HALO profile from right before the descent is also shown, with bars denoting measurement statistical uncertainty. The dashed green line shows the malfunctioning sonde profile.**


A much more statistically robust approach to comparing the DLH and HALO measurements is to utilize the near-field HALO measurements over the course of the campaign to DLH taken at flight altitude. Figure 8 and Fig. 4a show DLH WV
data from the four flights where it was operational, plotted at aircraft altitude with the HALO WV profiles below. The continuity of features between the DLH and HALO data in these plots confirms HALO's ability to capture UT/LS WV, plus the relatively moist environments of the ITCZ troposphere in Fig. 8c and the spiral descent in Fig. 4a. This agreement is quantified with the scatterplot in Fig. 9 and associated statistics in Table 2. Comparing the 86300 available DLH measurements with the nearest measurements in the coincident HALO profiles, ranging 0.004 g/kg to 2 g/kg, yielded an $R^2 =$
$0.66$ and a very small wet bias of 0.003 g/kg. A small wet bias could be expected due to the typical increase of WV with decreasing altitude, plus the range and distribution of the values in Fig. 9 suggest that this calculation may be dominated by the spatial variability over the ~400 m vertical separation between the measurements rather than indicative of a systematic bias in HALO. Overall, this good agreement of DLH and HALO WV with a large dataset spanning 3 orders of magnitude and a range of atmospheric conditions is a robust validation of HALO WV in the near-field, as well as the general
measurement principle and implementation.

**Figure 8. HALO WV curtains with DLH plotted at the aircraft altitude, as in Fig. 4a. Segments of three different flights are shown in (a, 18 April; b, 23 April) midlatitudes and (c, 26 April) the tropics. Only upper altitudes are shown to focus on comparison with DLH.**






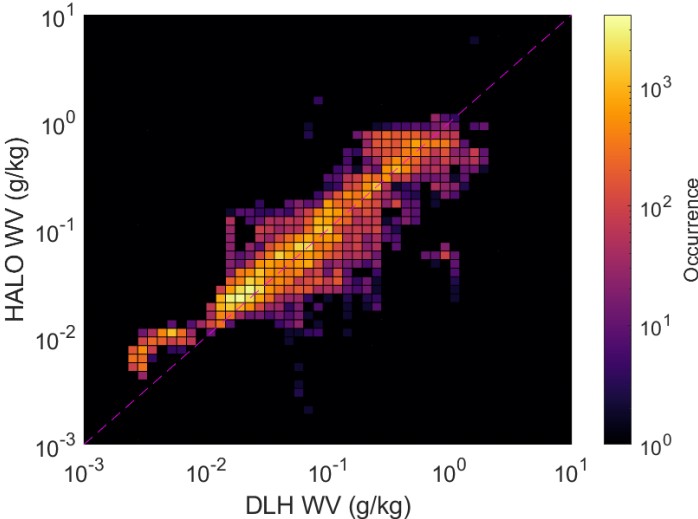

**Figure 9. Plot of DLH versus HALO WV on log scales and aggregated over all available data, with color indicating the number of points in each bin. The dashed line is a one-to-one line. HALO data is the average of the 5 highest-altitude measurements, ~400 m below the DLH.**

**Table 2. Statistics of HALO comparisons with other instrumentation. DLH comparison is against the average of the closest 5 HALO measurements. AIRS and IASI comparisons are of PWV, as shown in Fig. 11. *n* is the number of data points in the comparison.**

|  | *n* | Bias (HALO – x) | $R^2$ |
|---|---|---|---|
| DLH | 86300 | 0.003 g/kg | 0.66 |
| AIRS (ocean) | 44 | -2.55 mm | 0.96 |
| IASI (ocean) | 81 | -1.93 mm | 0.98 |
| IASI (land) | 24 | -4.07 mm | 0.85 |

### 5.3 Satellites

### 5.3.1 PWV

Geophysical observables such as WV profiles and PWV from satellites provide another opportunity by which to validate HALO against community standards (e.g., Chazette et al. 2014, Martins et al. 2010, Roman et al. 2016). During this campaign we had several under-flights of opportunity with AIRS and IASI that allowed direct comparisons to HALO WV profiles and PWV. HALO PWV is calculated by vertical integration of a given WV profile. It should be noted for the HALO and spaceborne sounder comparisons that HALO only gives a partial column PWV, limited to altitudes between the aircraft and surface (cloudy profiles have been omitted). PWV from HALO, AIRS, and IASI are shown on maps in Fig. 10




for comparison. Overall agreement is good, capturing the large moisture gradient near the ITCZ as well as features in
relatively dry midlatitudes. An important note for these comparisons is that some differences may arise from the brevity of
the satellite overpass (minutes or less) versus the hours of DC-8 flight that fall within the satellite swath. The DC-8 location
at overpass time is marked with a green circle in Fig. 10. This spatiotemporal offset may be responsible for some differences,
including the northwest corner of the oceanic portion of Fig. 10c, but cannot be corrected for in a straightforward manner.

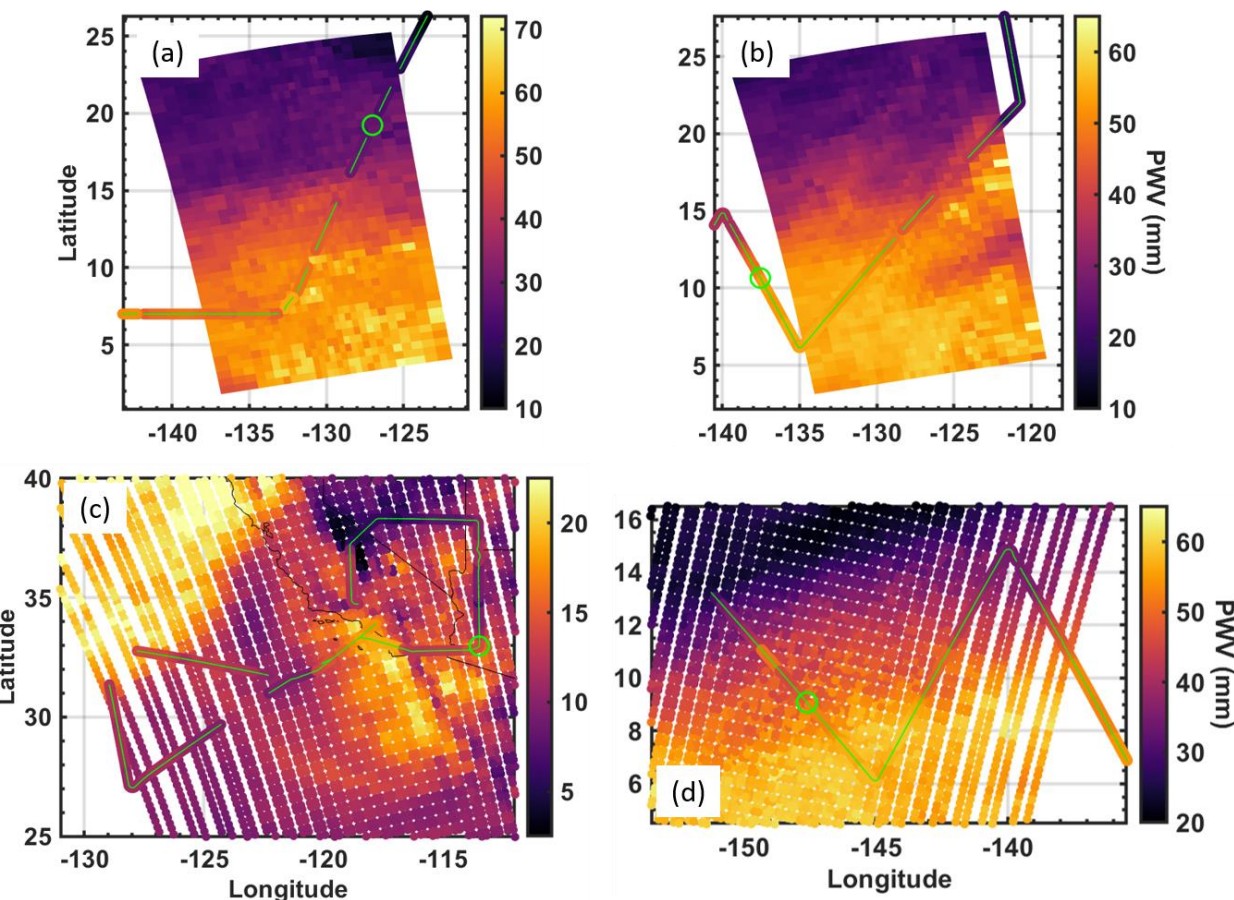

**Figure 10. (a, b) AIRS and (c, d) IASI PWV with HALO PWV overlaid. The HALO PWV has a green line at the center of the flight track for visibility. Green circles mark the time that the satellite overpassed. Gaps in the HALO data are due to clouds or missing data preventing full-column PWV retrieval. Dates are (a) 04/25, (b, d) 04/27, and (c) 04/23.**

Figure 11 and Table 2 show that there was excellent agreement of HALO PWV with AIRS and IASI, with HALO
measurements over ocean having a dry bias of 2.55 mm and 1.93 mm and $R^2$ of 0.96 and 0.98 against AIRS and IASI,
respectively. A dry bias is expected for these comparisons because HALO is only capturing the partial atmospheric column,
i.e., below the DC-8 flight altitude. The bias is most prevalent at high PWV values, which correspond to tropical flight legs



where the DC-8 was often flying within the upper reaches of deep ITCZ moisture plumes, e.g. Fig. 8c. On midlatitude flights
with low or moderate PWV and a relatively dry upper troposphere, such a bias was not clearly evident.

Some of the IASI overpass time included data over land (e.g., Fig. 10c), wherein the surface return DIAL technique was not
employed to measure the lowest few hundred meters of the atmosphere, and thus measured PWV is expected to have a larger
dry bias. The bias from this limitation is shown in Fig. 11 and Table 2 to be 2.14 mm larger than the PWV comparison over
ocean which included the surface return DIAL technique. This discrepancy should be eliminated in the future as the surface
return retrieval is applied to HALO data collected over land.

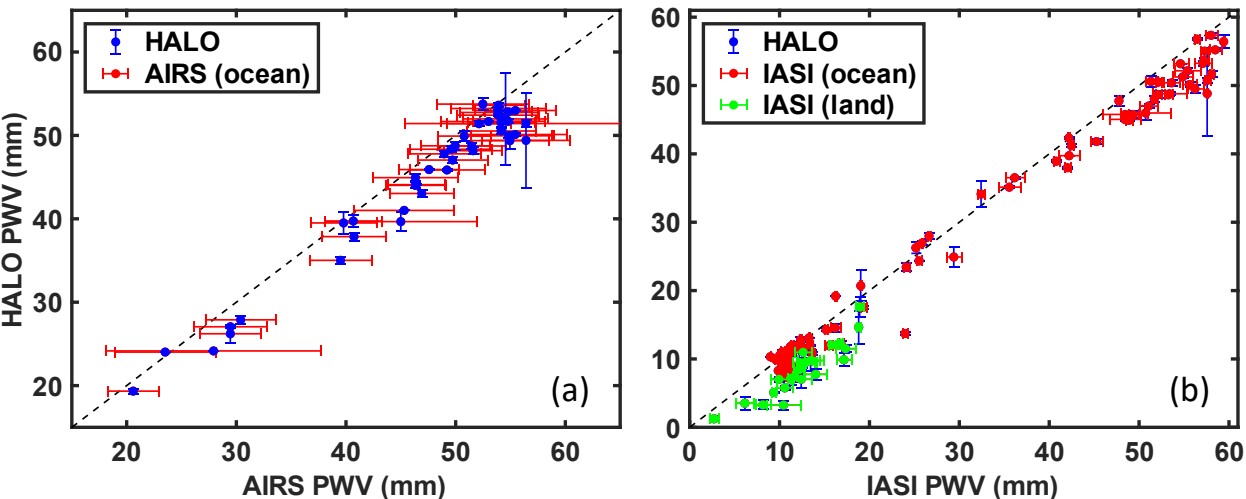

**Figure 11. Comparison of (a) AIRS or (b) IASI PWV with the average HALO PWV calculated within each satellite retrieval**
**footprint. Bars for AIRS are the reported retrieval uncertainties. Bars for HALO and IASI are standard deviations. Dashed line is**
**1:1, for reference.**

**5.3.2 Water vapor profiles**

Figure 12 shows four HALO WV profiles alongside AIRS and IASI WV profile products. The lowest AIRS point is the WV
mixing ratio retrieved at the surface, which has been appended to the atmospheric profile for this plot. As noted in Section
2E, the spaceborne sounders report at finely spaced vertical levels but are actually limited to 1 – 2 km vertical resolution of
independent information in the troposphere under ideal conditions. While areas of good agreement can be found, and some
differences may be attributed to spatiotemporal changes, the spaceborne sounders are ultimately very limited relative to
HALO in their ability to resolve sharp gradients and small-scale variability in moisture that are important for transport and
convection. This is especially apparent in these profiles around PBL top and tropospheric dry layers. The sounders capture
the general shape of the profile, at best.



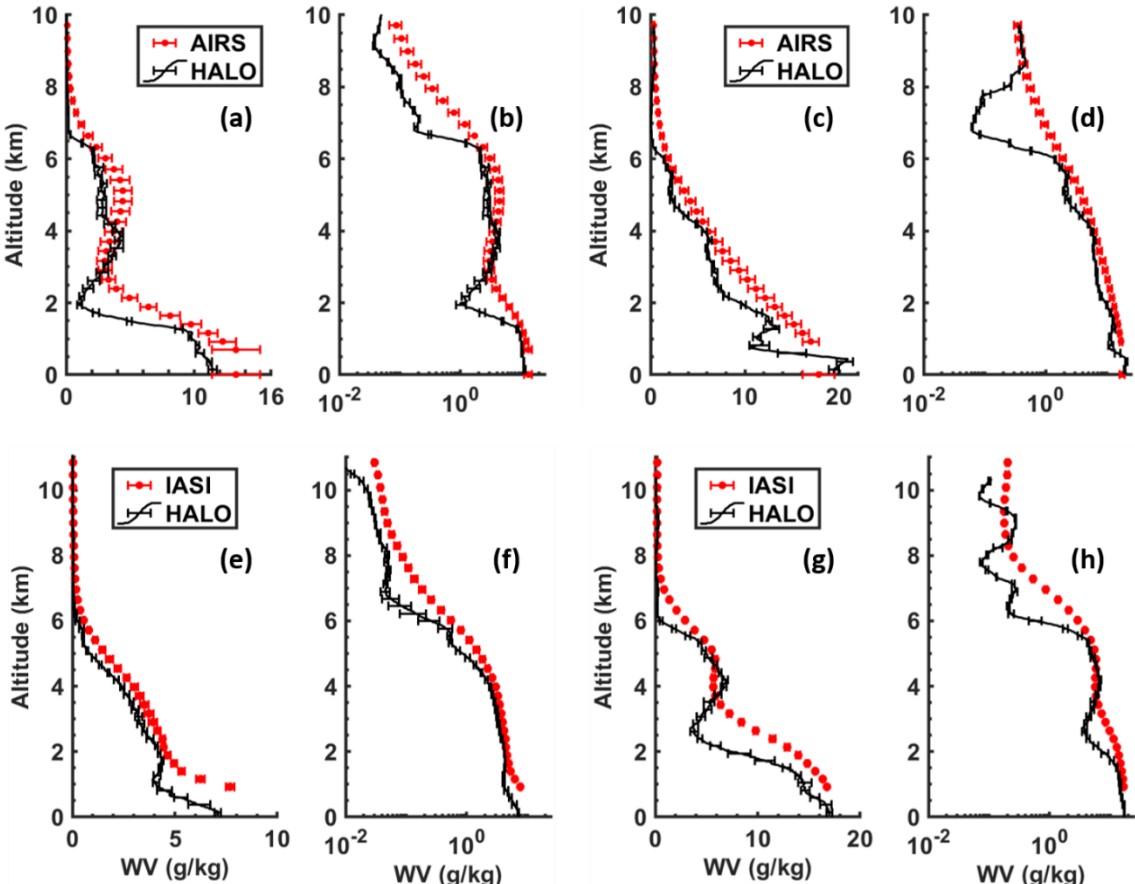

**Figure 12. WV mixing ratio profiles from HALO versus (a-d) AIRS and (e-h) IASI with two locations each and both (a, c, e, g) linear and (b, d, f, h) log scales to emphasize capabilities across scales. The AIRS profiles are from 25 April 2215 UTC overpass, with the HALO profiles from (a, b) 2236 and (c, d) 2316 UTC. IASI profiles are from (e, f) April 23 0400 UTC overpass and 0430 HALO profile, and (g, h) 27 April 1926 UTC overpass and 1912 UTC and HALO profile.**

AIRS and IASI provide important research and operational data to the community and have the strength of frequent global coverage, but HALO or similar active remote sensing is clearly advantageous for supporting process studies and other applications requiring higher spatial resolution and accuracy. The synergistic strengths of combining active and passive sounders will be an important resource moving forward, such as the work by Turner and Lohnert (2021) combining passive and active remote sensing observations to increase information content and improve vertical resolution and accuracy of passive retrievals.



**6 Conclusions**

This manuscript provides an overview of implementation and retrieval methodology for the new HALO airborne WV DIAL system and validation from its five maiden flights. The HALO instrument and WV retrieval were designed based on decades of legacy of related DIAL efforts at NASA LaRC and in the global community. The HALO WV DIAL measurements are carried out in the 935 nm spectral range, transmitting four wavelengths to achieve sensitivity to moisture from the UT/LS down to the PBL within a single profile. HALO is the first WV DIAL system to employ shot-by-shot switching between the

four wavelengths using a single laser transmitter, and in doing so reduces instrument size, complexity, and potential for certain systematic errors. The retrieval methodology incorporates the flexibility of the DIAL technique to trade resolution for precision, with streamlined reprocessing to optimize for scientific applications across scales. Another unique advantage of HALO is the combination of the WV DIAL with HSRL (or methane DIAL). The HSRL measurements provide cloud and aerosol optical properties, which are assets to both the WV retrieval calculations and scientific analysis.


The maiden flights of the HALO WV DIAL provided opportunities to validate the WV measurements in various forms over a wide range of atmospheric conditions spanning the tropical and midlatitude eastern Pacific. The DLH was operational onboard the DC-8 for most of the campaign, providing a large dataset of in situ WV observations for comparison to the HALO measurements nearest the aircraft (~400 m below). Values spanned 0.004 g/kg to 2 g/kg, and a very small bias of

only 0.003 g/kg was found which was likely dominated by spatial variability rather than systematic bias. There was also considerable spread with $R^2$=0.66, again due in part to spatial variability. A single DLH in situ profile was available for comparison to HALO profiles, and the DLH data was found to be within the range of values measured by HALO during the preceding overpass (with the exception of three very small deviations), indicating good agreement within the expected spatiotemporal variation for the area. Unfortunately, a typical assessment of profiles via statistical comparison to sondes was

not possible due to deficiencies in the sonde moisture sensors throughout the campaign. Despite this limitation, the HALO and sonde profiles were found to capture WV features and magnitudes with good agreement.

HALO PWV comparisons to community standard spaceborne sounders AIRS and IASI showed excellent agreement. PWV observations over ocean ranged 10 to 60 mm with $R^2$=0.96 and 0.98 for AIRS and IASI, respectively. HALO measurements

were often slightly drier than the sounders in accordance with expectations, considering HALO only measures a partial column based on DC-8 flight altitude. HALO profile comparisons to the sounders' WV profiles highlighted the different capabilities of the instruments, with HALO resolving much more detail and sharper gradients throughout the troposphere. Synergies between active and passive sounders such as these will be critical for improving measurements and information content on global scales in the future (Teixeira et al. 2021).




Although substantial conclusions were drawn, the opportunities for validation of HALO WV during the Aeolus cal/val campaign were not ideal due to the focus and brevity of the campaign. Further validation of HALO will be performed during future campaigns to quantify any potential systematic or other sources of error beyond the statistical uncertainty that is currently reported, though no such errors are evident to date. Improvements to the HALO architecture and processing are also ongoing, including efforts to expand the tunability of the transmitted wavelength targeting the lower troposphere, which will improve performance and expand measurement capability in very moist environments such as the ITCZ. Another major advancement will be the refining of the surface return DIAL algorithm over oceans and attempting to extend this capability to land as well as cloud top, which will further increase utility for PBL and cloud process studies in addition to column products such as PWV.

**Data availability**

Data from the Aeolus cal/val campaign is available through the NASA Atmospheric Science Data Center (ASDC) at https://doi.org/10.5067/SUBORBITAL/AEOLUSCALVAL2019/DATA001 (NASA/LARC/SD/ASDC).

**Author contribution**

Co-authors worked collaboratively on each aspect as part of the HALO team. ARN led conceptualization and development of the HALO instrument with contributions from AN and DBH. SK and JC led data curation and the HALO WV retrieval code with contributions from ARN and BJC. JC, JL, RAB contributed to the integration and operation of HALO during the campaign. BJC and ARN led the analyses presented here. BJC prepared the manuscript with contributions from co-authors.

**Competing interests**

The authors declare that they have no conflict of interest.

**Acknowledgements**

We acknowledge funding support from the NASA Headquarters Earth Science Division, the NASA Earth Science Technology Office, and NASA Langley Research Center. We thank the DC-8 team at the NASA Armstrong Flight Research Center and the National Suborbital Education and Research Center for their support of the Aeolus cal/val campaign. We thank Joshua Digangi and Glenn Diskin for providing the DLH data used in the preparation of this manuscript. We also thank Kris Bedka for his leadership during the Aeolus cal/val campaign and fruitful discussions regarding synergistic DIAL retrievals. BJC was supported by the NASA Postdoctoral Program.



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
