# Peer review of "Differential absorption lidar measurements of water vapor by the High Altitude Lidar Observatory (HALO): Retrieval framework and first results"

_Atmospheric Measurement Techniques, 2021_

## Author Comment (AC1)

Response to Anonymous Referee #1 Comment on amt-2021-229

The authors would like to thank both reviewers for their thoughtful and detailed feedback, after which the manuscript has undoubtedly improved. Reviewer comments below are in green, while responses are in black.

Addressing *"Detailed Comments"*:

Line 120: Because of issues with the dropsondes, quantitative water vapor profiles between the DIAL and radiosondes could not be done. This is a major weakness of this paper. Can the authors speak to future plans for DIAL/Sonde comparisons?

This comment aligns with remarks from reviewer #2 about the ability to adequately validate with the given datasets. We agree with the issues the reviewers raise, and to reflect this the end of the manuscript title will be changed from "validation" to "preliminary results". Verbiage throughout the text will be similarly updated.

Additional sonde comparisons yielding robust validation statistics will be forthcoming with data collected from the CPEX-AW campaign in the tropical Atlantic/Caribbean (Aug-Sept 2021) and the upcoming IMPACTS mission in the northeast U.S> (Jan-Feb 2022). Preliminary comparisons between HALO and dropsondes from the CPEX-AW campaign show favorable agreement, but cannot be shared publicly until spring 2022 when the campaign data are posted to a publicly available archive.

Line 185, the DIAL equation.
The references:
*A. Theopold and J. Bosenberg, "Differential Absorption Lidar Measurements of Atmospheric-Temperature Profiles - Theory and Experiment," J Atmos Ocean Tech 10, 165-179 (1993).*
*Bosenberg, "Ground-based differential absorption lidar for water-vapor and temperature profiling: methodology," Appl Optics 37, 3845-3860 (1998).*
developed a methodology for the DIAL technique that accounts for the Doppler-Broadened Rayleigh scattering. In this development, one can not simply use a modified cross section to account for the Doppler broaded molecular scattering. How do the authors justify using the DIAL equation with the modified cross section and how does that compare with the results from the above two references?

We do employ the methodology outlined in those papers. It is apparent from both reviewer's comments that the original description of the approach in this manuscript was unclear, so the Doppler correction section has been rewritten.

As a side note, challenges faced in the Theopold & Bosenberg reference drew the conclusion that "high quality information on the different contributions to the backscattered signal is required," namely the contributions of particulate versus molecular backscatter. This is a distinct advantage of HALO having HSRL + DIAL, as noted in the manuscript.

Line 204  What is the physical reason for the 1.5 exponent in the delta r term?

As derived from Poisson statistics, random uncorrelated noise can be described by sqrt(n) where n is the number of received photons (or in the case of analog systems, number of digitizer counts multiplied by the noise scale factor). Increasing the number of along track accumulated shots or the vertical range bin size increases the received counts linearly and decreases the noise in the observed water vapor retrieval by $\Delta t^{-.5}$ and $\Delta r^{-.5}$, respectively. The additional 1.0 for the $\Delta r$ term is present because the DIAL technique relies on along-path optical depth where the random noise decreases linearly with accumulated optical depth. This $\Delta r$ originates in the extinction term of the lidar equation (second term in equation equation 2

from Nehrir et al. 2009, JTECH) and persists through to DIAL uncertainty, as shown in Ismail & Browell 1989. This reference has been made more clear in the text.

Line 240:  How was the random error plot generated i.e. what random errors are included here.
The following sentence has been added to the text: "Fig. 1c is a manually drawn estimate of the DAOD and Poisson statistics contributions (shot noise) to the WV uncertainty, meant to illustrate the typical sensitivity of each wavelength pair."

Line 264:  If detection is limited to 15 m due to the electronics, how can you get a 1.5 m resolution?
"Signals" in that sentence has been changed to "detection chain" to help clarify.
The detection chain for the WV DIAL is separate from the detection chain for HSRL. As discussed earlier in that paragraph, the 15m limit is for the WV DIAL detection chain.

Line 284:  Having the instrument paper would be a useful compliment to this paper.  I am not sure about the reference here to an as yet unpublished paper.  Could you say more about the status of this instrument paper?
The instrument paper is planned to be written this winter, but as it is still months away from the public forum, mentions of this paper have been trimmed in the text and no specific authors are named.

Line 314:  Please add a reference for self-broadening calculations.
This is not a special consideration, since a proper broadening calculation should consider every molecule in the atmosphere. It just so happens that our species of interest (WV) can make up to ~3-5% of the atmosphere, and is thus a non-negligible trace gas. Reference to the general broadening discussions of Ismail & Browell (1989) has been added and the text has been slightly reworded for clarity.

Line 379:  How do the choice of the OD theshold values affect the retrieval?  Will non-experts in lidar be able to do this manual inspection and can this be automated?
The OD choice does not need to be extremely precise, since the sensitivity ranges of the HALO wavelengths typically overlap as shown with Fig. 1. If an OD value is chosen that is too high or too low, it will typically be very apparent in manual inspection as insufficient DAOD in one of the wavelength pairs results in a very noisy retrieval. HALO WV profiles are archived with this splicing already done, so end users of the data don't need to do it. As for a non-expert in lidar would be capable of, that is hard to comment on as it would vary based on scene and relevant experience…
The following sentence has been added to the text: "Results are typically not affected by threshold changes within ±0.1 of chosen values because the overlap region where both wavelength pairs perform well is sufficiently broad."

Line 413:  Do you have any evidence that using the surface reflection over land or the reflection for cloud tops can be used to retrieve the water vapor mixing ratio in the lowest range bin?   With what accaurcy can you expect to retrieve water vapor inthe lowest range bin over land or cloud top?
The same principle holds true for making the retrieval over land/cloud as water, though as the text denotes, "Topographic and albedo variability of cloud and land will require a more detailed treatment to ensure surface-related changes in signal are separated from atmospheric OD variation." We have not done enough work on this additional capability yet to make a quantitative projection on performance. Text has been added to that paragraph to be more conservative about future efforts and results, since sufficiently complex surfaces/scenes may prohibit an accurate retrieval.

Line 436: How do errors in the assumed temerpature profile affect the accuarcy of the Doppler correction factor and overall water vapor retrieval?

The magnitude of the Doppler corrections to the HALO WV profiles from the Aeolus Cal/Val campaign datasets were 3.5% at most (this has been added to the text), with the largest magnitudes in the UT/LS and generally <2% in the lower troposphere.

We performed a sensitivity test by changing MERRA-2 temperature by ±10 K and found roughly 10% changes to the Doppler correction factor, at most. Because this is a small perturbation to an already small Doppler correction, and errors in MERRA-2 should generally be much less than 10 K, we neglect any compounding uncertainty that this may cause.

Line 475: When reporting the accuacy of the water vapor retrieval near the surace of less than 15% -- was this only over water?  If so please state.

Yes. This has been added to the text.

Unfortantely, the issue with the dropsondes results in the validation consisting of one profile comparison with the DLH profile that resulted from a spiral desent of the aircraft.  The comparison of the first range bin of the HALO with the DHL and the comparison of the pericitable water vapor show good agreement, but comparison with the DHL and HALO have a 400 m offset, coparison with the pwv from satellite based measurements will result in a dry bias for the HALO instrument since it misses the pwv above the aircraft, and comparsion of the satellite profiles and HALO are qualitative at best due to the spatial reolution of the passive sensor.  This results in a non ideal set of measurments for validation.  While this paper is still publishable and, I feel, should be published, I hope to see follow on validation efforts based on sonde profiles.

We share the reviewer's perspective on this and will take advantage of future datasets for a more rigorous validation of HALO WV. As mentioned above, the manuscript title and some related phrasing within the text have been changed to avoid conclusive "validation" terminology.

---

## Author Comment (AC2)

Response to Anonymous Referee #2 Comment on amt-2021-229

The authors thank both reviewers for their thoughtful and detailed feedback, after which the manuscript has undoubtedly improved. Reviewer comments below are in green, while responses are in black.

*Summary:*
The submitted manuscript analyzes the performance of the NASA HALO lidar system in its water vapor and HSRL configuration during 5 Aeolus calval flights from 2019. The lidar system is described in some detail with the processing steps described in more detail than the lidar hardware. Data comparisons are presented where possible including range resolved comparisons to dropsondes, an in-situ diode laser hygrometer, AIRS and IASI, with passive column measurements also compared to AIRS and IASI as well. WV data is presented covering 3 orders of magnitude, which is impressive from any sensor, but especially from a DIAL sensor that must be mounted on an aircraft. Detail related to this new advanced system is certainly relevant within the scope of AMT.

Overall, the analysis methodology presented is reasonable, albeit with flawed ancillary data, and the quality of the work is high. The explanations given are mostly clear, though I have suggestions below for areas of my personal confusion. The caveats the authors give that are used to qualify and describe the extent of the statements given are very refreshing and in my personal opinion rather brave, i.e. "we understand the limits of our analysis are here and they are not what we would have hoped".

However, as a reviewer, I see three overarching issues that can not be ignored, upon which my below comments will expand. First and foremost, the quantity and quality of the ancillary data used for validation does not give me confidence as a reviewer that this paper provides a true validation of the HALO instrument in the presented configuration. This is a major weakness that I believe disqualifies the presented data set from being used for validation without substantial additional information. Second, the retrieval framework is presented with hardly any detail related to specific error sources and magnitudes; this makes it very difficult to evaluate this new system within the context of previously described lidar systems, or indeed other non-lidar sensors. Basically, I find it very difficult to judge if the results observed are "The right answer for the right reasons". Third, it seems the authors are trying break up a large amount of analysis and data over a few manuscripts (an instrument paper, a validation, and a description of a flight campaign, i.e. Bedka et al. 2021). While this is reasonable in principle, the execution leaves some areas of importance unexplored and leaves the interested reader to have to dig through multiple resources containing somewhat redundant information looking for details. This seems to be rather like threading a needle of detail. There is significant overlap of conclusions, comparisons, and information with Bedka et al. 2021 in particular. This seems to be necessitated by publishing a campaign description before the validation before the instrument paper. This significant overlap of information is detrimental to the impact of this paper by itself. Said more concisely: this paper is not, nor does it seem to be intended to be, a definitive resource. Furthermore, it promises more work to be presented in the future related precisely to the scope of the presented manuscript, which is not really admissible evidence in the context of assessing the overall merit of the presented manuscript.

I would suggest that major revisions are required before publication of this work. In truth, perhaps "major additions" are what is needed, in my opinion, as the data and methodologies presented seem perfectly reasonable. It seems that the authors have done about the most they can given the highly limiting constraints of the chosen ancillary data set. HALO data is analyzed with an incomplete error description applied to a flawed data set that causes too many caveats and externalities to be ignored. I have broken

my comments into major and minor comments as well as suggestions which should be understood as very minor comments.

*Major Comments:*

1. If I may bluntly summarize the data set used for validation, it consists of a set of untrustworthy dropsondes, a single vertical profile from the DC 8, in situ comparisons of the DIAL system's first range bin, and satellite data that is coarsely range resolved in the region of interest. Taking none of the below comments into account, I question very strongly whether that is sufficient for a validation. It seems the authors had hoped the ancillary data set would be more conclusive, but were left with an impossible task of salvaging poor data. I applaud the author's honestly with lines 631-635: "Although substantial conclusions were drawn, the opportunities for validation of HALO WV during the Aeolus cal/val campaign were not ideal due to the focus and brevity of the campaign. Further validation of HALO will be performed during future campaigns to quantify any potential systematic or other sources of error beyond the statistical uncertainty that is currently reported, though no such errors are evident to date." In my opinion, this is disqualifying of the draft in its current form without the future data, which is not yet available. I do not think the promise of further validation efforts is admissible in this case as the work is not done nor presented. These future efforts are precisely within the scope of this manuscript. It is effectively assumed here than no further issues are present. To the contrary, should it be assumed that there is an error in the system that is as yet undetected, having a single validation in the publication record that did not well cover the system's deployment potential is both confusing and risks being contradicted.

   We agree with the reviewer and have changed text in the title from "validation" to "preliminary results", as well as changing most "validation" verbiage in the text to more conservative statements. These were flights of opportunity for HALO without optimal sampling strategies. While further validation will come in future publications, this does not disqualify the utility or impact of this manuscript. This manuscript will largely serve to inform the community of a new asset and capabilities in the airborne portfolio, communicating relevant information for HALO to be optimally leveraged in future field campaigns.

2. I am surprised at the lack of treatment of retrieval errors. I would argue that analysis of all known systematic and random errors is very much in scope of a paper claiming validation of an instrument. This is especially true where the retrieval methodology contains a significant amount of flexibility for human intervention and interpretation. This flexibility is emphasized in this manuscript. Only statistical errors are presented in my reading of this manuscript. The comment from Lines 201-202 about future error discussion given by Nehrir et al. is not sufficient, especially as error bars and error data are extensively presented as in Figures 3, 4, 6, 11, and 12. Relevant for data from, for example Figure 4, where single digit uncertainty is reported on water vapor quantities less than 0.03 or 0.04 g/kg; this is both potentially a demonstration of a disruptive technology and also potentially misrepresenting the underlying data. I would expect multiple single elements of a full error budget to be in excess of that amount at that level of precision/accuracy. I agree with the authors that much of the theoretical groundwork for general DIAL based errors has been laid and does not need to be rehashed. However, the errors presented resulting from this retrieval framework specific to HALO have missed an enormous amount of detail that is critical for anyone seeking to use this data in the future or anyone trying to evaluate HALO data in the context of non-lidar sensors. Some examples are:

   As stated in the text, the uncertainties plotted here are only considering Poissonian statistical uncertainty. The reviewer is correct that other potential sources of error need to be considered,

and they have, but are largely negligible or already accounted for. In addition to the specific points addressed below, the following sentence has been added to the text: "Additional sources of systematic uncertainty such as errors in HITRAN, knowledge of atmospheric state (namely temperature), or propagating errors from HSRL products used in Doppler correction are expected to be <1% and not accounted for here. Additionally, the magnitude of systematic errors resulting from uncertainties in the performance of the instrument such as knowledge of the transmitted wavelength, spectral purity, or linearity of the receiver detector chain are estimated to be <2%, far below the magnitude of the random error resulting from detector electronic noise and shot noise. These sources of systematic error account for a larger fraction of the error budget in drier environments such as in the UT/LS, however, the comparison with DLH demonstrate that systematic sources of error are to a large extent well understood and don't drive the overall error budget of the retrieval."

Treatment of sources of systematic error resulting from the instrument will be discussed in the instrument paper.

1. How do you balance averaging, thereby reducing statistical error, with increasing the range of observation, thereby increasing your error in representativeness? Line 331-332 suggest there is no consideration to uncertainties from data smearing. See for example Hayman et al. 2020 on this topic.

   This is indeed an unquantified error as is the case for any instrument that requires averaging to overcome random noise. Given the DIAL retrieval is flexible in trading spatial resolution for precision, it is conceivable that the spatial resolution could be increased to capture the highest frequency variability in the observed quantity (e.g. water vapor mixing ratio), however, this is well beyond the scope of this paper as optimization of the spatial resolution involves working with end user on an application specific basis. Work like Hayman et al. (2020) tries to minimize this potential source of error of a low SNR retrieval which requires substantial temporal averaging, but a key difference for their work is using a ground-based system where the sensed volume is relatively static. The averaging windows required for an airborne system to produce useful retrievals are long enough to potentially include much more spatiotemporal variability, and additional research would likely be needed to assess viability and optimize similar techniques. This may be a future effort of our group but is not logistically feasible to tackle right now. In scientific applications of this data we make it clear that manual inspection and reprocessing is available to optimize for various science targets. Comments have been added to the text to reinforce the point that averaging reduces statistical uncertainty but may introduce errors in representativeness.

2. Presumably HSRL data come with an error estimate. That error is not described here at all as far as I can tell. How do those errors propagate into your Rayleigh-Doppler corrections?

   As described in Hair et al. 2008, the combined systematic error in the aerosol backscattering coefficient associated with the gain calibrations, iodine filter transmission, laser spectral purity, atmospheric state parameters, and molecular depolarization is estimated to be less than 3%. Given the HSRL architecture employed in HALO is the same as the HSRL instrument described in Hair et al. 2008 and the knowledge of the performance of the relevant HALO transmitter and receiver subsystems are known to high accuracy, we assume the systematic errors in the HALO backscatter observations are of the same magnitude of those discussed in Hair et al. 2008. The random errors resulting from shot noise variability for the HALO aerosol backscatter (for the same spatial resolution) are also on order of those presented in Hair et al. 2008 and assumed to be less

than 5%. The HALO extinction products have been validated in Wu et al. 2021, the precision and accuracy of which are similar to those reported in Hair et al. 2008.

All of that to say, the random and systematic errors in the retrieved backscatter are small, and such a small uncertainty applied to an already small Doppler Correction term (<~3.5%) would result in negligible changes to in the Doppler correction term in the DIAL retrieval. As such, we do not propagate uncertainties in the aerosol backscatter through to the Doppler correction term.

3. Angstrom exponents are used to convert HSRL data to the 935 nm wavelength. Is that an exact number or range and does the uncertainty in that exponent cause further uncertainties?

The use of angstrom exponent to obtain backscatter at a specified wavelength is inherently an estimation but the error introduced cannot be known without additional information. However, because the Doppler broadening correction that uses the angstrom exponent is so small (~3.5% at maximum), propagating error from the angstrom exponent would be an even lesser-order effect and is neglected here. The following text has been added to better describe the angstrom exponent methodology: "the aerosol backscatter profile at 935 nm is estimated from the backscatter angstrom exponent at 532 nm / 1064 nm (Burton et al. 2012). This angstrom exponent comes from the HSRL 532 nm aerosol backscatter and the 1064 nm aerosol backscatter retrieved using Equation 10 of Hair et al. (2008). If there are brief periods when DIAL data is available but HSRL is not (e.g., 3 UTC in Fig. 4), the last available HSRL profile is used."

4. Are there errors in calculating the Noise Scale Factor or is that assumed to be known/retrieved exactly?

The NSF is retrieved for a set of system parameters, which we try to keep constant throughout the campaign. Some potential challenges in NSF application are discussed in Liu et al. (2006). Further advancement of NSF theory is beyond the scope of this work. The following sentences have been added to the text: "A mean NSF is used for each channel in each campaign, calculated from the NSFs determined from each flight. Because relevant instrument parameters are kept constant to the greatest extent possible, the NSFs are fairly constant over the campaign, with standard deviations from the campaign mean <15%."

5. What are your assumed spectroscopic errors based on the Hitran 2016 database?

We have not calculated the contribution to uncertainty from HITRAN. This is not a current focus area for our team, since HITRAN exists as a resource to circumvent this type of work and the community has already provided focused studies of the 935 nm region for contribution to HITRAN (e.g., the following references). If/when this team transitions to a space-based mission, funding devoted to additional spectroscopic activities would be a priority. Based on the work carried out in the references below, we estimate a systematic error of <1% due to uncertainty in the knowledge of the absorption cross-section. Future efforts could (will) focus on different treatments and second order effects (e.g. speed dependance and line mixing) of the absorption line shape.

- Birk, M., Wagner, G., Loos, J., Lodi, L., Polyansky, O. L., Kyuberis, A. A., ... & Tennyson, J. (2017). Accurate line intensities for water transitions in the infrared: comparison of theory and experiment. *Journal of Quantitative Spectroscopy and Radiative Transfer*, *203*, 88-102.
- Hodges, J. T., Lisak, D., Lavrentieva, N., Bykov, A., Sinitsa, L., Tennyson, J., ... & Tolchenov, R. N. (2008). Comparison between theoretical calculations and highresolution measurements of pressure broadening for near-infrared water spectra. *Journal of Molecular Spectroscopy*, *249*(2), 86-94.

3. It seems like this instrument really requires monotonically, or nearly monotonically, increasing water vapor content with observed range. In principle with a canonical water vapor distribution, this is reasonable. However with a non-standard distribution, this utility might be affected. If for example a moist layer sits on top of an extremely dry layer, the sensitive DIAL pair $\lambda_1$ and $\lambda_2$ would be extinguished before its data was useful. It is in my opinion therefore crucial to examine cases where water vapor profiles do not follow a canonical increasing trend with range from the instrument. This comment comes with a few questions:

Replying to the paragraph above and sub-questions below, as they are all related:

It is an inaccurate statement to say that the instrument requires monotonically increasing water vapor content in range to employ the presented retrieval framework. The manuscript provides several example scenes with dry layers in the mid/low troposphere bracketed by moist layers on either side, as can be seen in the curtain plots from the full Aeolus Cal/Val campaign shown in Bedka et al. 2021 and Fig.s 3 and 4 here. The challenge that the reviewer mentions here of sensitivity to distant dry layers is not unique to HALO but true of the DIAL technique in general. More absorbing wavelengths will always be attenuated closer to the instrument and thus a flexible wavelength selection or tuning architecture, as with HALO, is the only way to mitigate these optical depth challenges (not considering more laser output power or other hardware considerations that could be employed to improve performance). During operation, the HALO $\lambda_3$ signal is monitored and tuned as necessary such that the received signal stays above the instrument noise floor. This effectively maintains maximum possible sensitivity to any dry layers that may be present, while still capturing the full profile to the surface. That being said, there is inevitably low measurement precision in dry layers when a less absorbing online wavelengths is employed, such as those presented in the lower troposphere in this manuscript.

A brief paragraph to this effect has been added to Section 4.2.

   1. Are extremely dry layers between moist layers expected to be observed well?
   2. Does your processing method of assembling a profile based on decreasing moisture absorption with increasing range from the instrument fundamentally limit your ability to discover unusual atmospheric situations that do not follow this canonical form?
   3. Taking as an example Figure 4, how sensitive to the extremely dry layer should your least sensitive DIAL wavelength pair be? Should this data simply be flagged out or trusted, or simply have massive error associated with it?

      *Building on the reply above:* This measurement should not be flagged out because we still have good online signal strength in this layer but just a very low DAOD, and are thus confident that there is a very dry layer despite a loss of measurement precision. This is valuable information to retain, and anyone using the HALO WV datasets should be in contact with the instrument team to help with interpreting such scenarios.
   4. Is the dry layer observed by the dropsonde in Figure 6 panel d real? It appears that there is a lack of evidence to refute it other than a general misbehavior of the sondes (though that sonde comparison otherwise looks pretty good if you do a mental shift of the data in range).

      We have no reason to believe that the dry layer in Figure 6d isn't real. The poor correlation between the sonde and HALO could result from a mismatch in the atmospheric sampling volume. A more statistically significant validation data set in the future will help identify the extent to which the instrument can accurately retrieve similar mid-altitude dry layers.

4. Section 2 feels out of place and repetitive. In my opinion, it is difficult to understand what is going on and why it is being presented before the HALO system and capabilities are described. Furthermore, I believe too many comparison-specific details are presented in section 2, causing repetition. I would suggest that the following might improve the readability of this manuscript.
    1. A more logical location in my opinion for the current section 2 is between the current sections 4 and 5.
    2. Data specific to the nature of comparisons (lines 116-120 for the Dropsondes, lines 132-141 for the DHL, lines 156-164 for AIRS, and lines 176-182 for IASI) should probably be moved to Section 5 to avoid repetition.

The layout of the text followed general convention to describe all instruments and measurement techniques prior to jumping into specifics. The end of section 1 provides the reader insight into the layout of the paper and context for including section 2 prior to detailed investigation of the DIAL retrieval and associated results. We compromised with the reviewer by moving remarks specific to the nature of comparisons to Section 5, and left instrument-specific information in Section 2.

5. Lines 235-238: Regarding the use of sequential wavelengths as DIAL wavelength pairs, I am not sure the reasoning presented completely follows. It is clear that you are using the most closely spaced pairs optimizing your assumptions vis-à-vis scattering cross sections and extinction. However, my guess is it places unnecessarily limits the analysis you can perform. Does an increase of wavelength separation of 0.2 nm quantitatively impact your error budget in any significant way?

The reviewer is correct, a separation of ~0.2 nm will not impact the error budget in a significant way, however, there are second order effects that do push us to utilize sequential wavelengths in our retrieval rather than referencing all online wavelengths to lambda 4.  One motivating factor is to minimize the angular dependence of the near-field returns through the narrowband optical filter (NBF).  Nehrir et al. 2009 demonstrated that differential wavelength shifts through the NBF can result in systematic bias in the nearfield.  To overcome this HALO has employed spectrally flat narrowband filters, however, we have seen evidence of nearfield bias when using wavelength 4 as the DIAL reference. This implies there are still small changes in the filter transmission between wavelengths and that minimizing spectral separation helps to improve retrievals in the near-field. This point has been added to the text.

Furthermore, at 200 m/s flight speed and 1000 Hz laser rep rate, you are talking about 20 cm between shots. It shouldn't matter if you take online or offline data first or second, so the laser shots should not be further than 40 cm apart in any combination. If you do data analysis at the 2 Hz rate of DAQ reporting or if you do analysis with the multi-second averaging you report, the time between shots should be negligible. We agree that this will likely never be a significant difference for the HALO architecture and have removed this motivating point from the text. With that said, we still believe that attempting to minimize spectral separation is beneficial to overcome systematic errors in heterogenous scenes (e.g. cloud edges, surface retrievals over land…etc.) if it doesn't impose risk or additional challenges to the hardware implementation, which is not the case in the HALO system.

I do not understand how it matters at all what order you take data in. However, I do see a couple of significant reasons to use a single laser wavelength as the offline laser including minimizing Rayleigh-Doppler broadening and maximizing available signal. Figure 1b exemplifies that "maximizing available signal" is not worthy of concern. Rayleigh-Doppler broadening for $\lambda_1/\lambda_4$ versus $\lambda_1/\lambda_2$ only affects a change <0.5% in the WV product. These two points have been added to the text. This seems to be suggested for example in Figure 1 of Nehrir et al. 2017, albeit with a

different offline wavelength. I believe this analysis needs to be expanded here to at least address the following:'

1. What effect, if any, does picking a different wavelength for the offline channel have on the results reported? Said differently, should you not get the same answer aside from statistical noise with any available choice of offline wavelength given the same online wavelength? Here I mean specifically is data from $\lambda_1$ and $\lambda_2$ substantially different than would result from using $\lambda_1$ and $\lambda_3$ or $\lambda_1$ and $\lambda_4$ and is $\lambda_2$ and $\lambda_3$ substantially different than $\lambda_2$ and $\lambda_4$?

   As discussed above, there are no notable differences in the retrieval from a theoretical perspective. In practice, as long as the differential extinction (contrast) between the 'on' and 'off' wavelength are sufficiently large, the selection of the offline reference wavelength is not critical. We employ minimal spectral separation to 1) overcome nearfield systematic effects discussed above and 2) employ best practices when possible to optimize retrievals in heterogenous conditions. As also discussed above, our approach of utilizing subsequent wavelengths for on/off DIAL pairs isn't an indication that other approaches couldn't or shouldn't be used, it's just the method we have employed and presented in the manuscript.

2. Is the difference in measured water vapor from different DIAL wavelength pairs from question 5a indicative of or help quantify any error?

   Yes, this is a very good point and a current subject of investigation by our group. As discussed above, nearfield retrievals yield different results depending on the spectral separation of the on/off pairs. Utilizing non-optimized DIAL pairs (i.e. utilizing wavelength pair 2/3 in the upper troposphere and 3/4 in the middle troposphere) should result in the same answer as for the 'optimized' retrieval, however, because the differential cross-section is not optimized (i.e., too small for a given water vapor density), any sources of systematic error such as non-uniform or time-variant electronic baseline (unique to analog systems) would dominate the retrieval error budget. These sources of error are negligible when the 'optimized' DIAL pair is employed for the correct part of the atmosphere.

3. Does the choice of using sequential descending wavelength sensitivities vs. using a single offline wavelength have any negative consequences?

   *We do not see this question as anything unique from the adjacent comments and responses.*

4. Does the size of the Rayleigh-Doppler correction increase substantially for $\lambda_2$ or $\lambda_3$ as offline channels vs. $\lambda_4$? $\lambda_4$, by sitting on the side of a line and not on top, would seem to me to be the least sensitive to Rayleigh-Doppler broadening.

   The reviewer is correct that $\lambda_4$ is least sensitive to Rayleigh-Doppler broadening. The Doppler correction to the final WV product was 3.5% at most for the entire campaign, maximized in the UT/LS. We calculated a change of ~6-9% in the Doppler correction when using $\lambda_1/\lambda_4$ versus $\lambda_1/\lambda_2$ in the UT/LS. This is thus a sufficiently small difference to an already small correction term to the WV product that we choose to neglect it.

6. It is unclear to me how you handle data dropouts from the HSRL (for example roughly 3 UTC on Figure 4). If you need that data to perform spectroscopic corrections, how do you handle that lack of data? Does this lack of data reflect in increasing error quantities? I would think some mention of the way this is handled is necessary.

The following sentence has been added to the Doppler broadening section: "If there are brief periods when DIAL data is available but HSRL is not (e.g., 3 UTC in Fig. 4), the last available HSRL profile is used."

This method was tested against other straightforward approaches (e.g., using an average HSRL profile across a leg or whole flight) and found to work well.

Because the magnitude of the Doppler correction is so small (<3%), further discussion of compounding uncertainties in this case are not within the scope of this paper.

7. What order of magnitude are the applied Doppler corrections? In particular with your extremely dry data set, I would think it should be very sensitive to Rayleigh-Doppler broadening as both wavelengths are parked on the peak of an absorption feature. In the manuscript the fact that this correction is applied is given as a matter of fact but the magnitude is likely important, i.e., as an example with made up magnitude, if it is a 2% correction it is no worse than statistical noise, 20% would be worth knowing, and 200% would possibly mean it should be flagged as bad and ignored. The work of Späth et al. 2020 might be a useful reference here.

The magnitude of the Doppler corrections to the HALO WV profiles from the Aeolus Cal/Val campaign datasets were 3.5% at most (this has been added to the text), with the largest magnitudes in the UT/LS and generally <2% in the lower troposphere. This is within a factor of 2 of the optimal scenarios presented by Spath et al. 2020.

8. Section 4.3 seems incomplete in a number of respects. You claim in the abstract that this IPDA-type technique is presented. However IPDA is not novel per se as is referenced to the work of Barton-Grimley et al. 2021 among others. There is also no data presented from the test flights in this section other than comparisons to dropsondes, which have been noted to be unreliable, and to satellite sensors, which have been noted to have poor measurements near the ground. Finally, without data presented, there are a number of claims of empirically determined processing steps that are supported by vague statements that don't seem supportable.

- As the text indicates, this is distinct from IDPA (a column measurement) as it yields a range-resolved product and is optimized for the lowest part of the atmosphere.
- We agree with the reviewer that thorough validation is not possible with the current dataset, hence the tempering of the related language throughout the manuscript (as already mentioned for a previous comment.)
- There is more data presented than the reviewer suggests, including the DLH comparison in Figure 7 and the WV curtain in Figure 4, as well as the contribution to PWV calculation. The WV curtains in Bedka et al. (2021) also contain the retrieved values for the full campaign. Again, the opportunities for quantitative validation were very limited, but the near-surface product was developed to minimize physically implausible values (e.g. sudden large changes in WV without apparent cause) across the entire dataset and agree with the available comparisons. This paper is mainly meant to inform the community of a new instrument/capability within the airborne portfolio, and we are confident that this is a viable technique to broadcast within that framework.
- Regarding comparison to spaceborne sounders, the following sentence has been added to the PWV section: "The overall good agreement may also be interpreted as an indirect indicator of accuracy in the near-surface retrieval presented in Section 4.3, since PWV is generally dominated by the high moisture content of the PBL (e.g., Richardson et al. 2021, Thompson et al. 2021), but this is not a direct relationship appropriate for robust validation."

- Thompson, D. R., Kahn, B. H., Brodrick, P. G., Lebsock, M. D., Richardson, M., & Green, R. O. (2021a). Spectroscopic imaging of sub-kilometer spatial structure in lower-tropospheric water vapor. *Atmospheric Measurement Techniques*, *14*(4), 2827-2840.
- Richardson, M. T., Thompson, D. R., Kurowski, M. J., & Lebsock, M. D. (2021b). Boundary layer water vapour statistics from high-spatial-resolution spaceborne imaging spectroscopy. *Atmospheric Measurement Techniques Discussions*, 1-33.

1. Lines 398-400: Optimal signal strength is claimed with no data presented over urban or rural land or ice. Presumably they are different but what then is optimal. Does it change flight to flight or hour to hour?

   "Optimal" is already defined within that line as "ensuring that the strong signal is captured within the linear regime of the channel's digitizer," but to further clarify the following parenthetical has been added: "(i.e., neither saturating nor reaching the noise floor.)"

   As the reviewer suggests, this gain channel choice does vary depending on scene and viewing geometry. Preliminary analysis of data from the recent CPEX-AW field campaign has shown the occasional need for changing which channel is used for the surface return depending on aircraft altitude, near surface wind speed, and aerosol/cloud optical depth between the aircraft and surface. This sentence has been added to the text: "The low-optical high-electrical gain channel consistently best captured the surface return signal over ocean during this campaign, but this channel choice may differ for other datasets, e.g., over ice or flying at lower altitudes."

2. Lines 402-403: It is unclear what you mean by improvement here as you have not defined a reference measurement or error as a comparison method.

   This sentence has been removed.

3. Lines 404-405: Empirical methods to reduce outliers seems rather heavy handed as no data is presented to prove an outlier is not valid. By what criteria are outliers identified?

   The development of this retrieval code leveraged the few comparison measurements that were available and then largely relied on non-physical results to rule out bad approaches, e.g., integrating fewer than 5 points for the surface return produced unreasonably high variability and magnitudes in the water vapor field. The retrieval principle is theoretically sound, and such choices in the retrieval were fairly straightforward for producing or avoiding large errors. Small changes have been made to the text of this section.

4. Lines 474-476: Here a claim of validation of the surface result is achieved with sondes that have potentially several hundred meter altitude offsets to a return calculated with the last approximately 100 meters of HALO data?

   The validation claim has been removed.

9. Lines 452-454: This section suggests to me as a reader that detailed analysis of the issues experiences with moisture from the dropsondes can be found in Bedka et al. 2021. However, that manuscript says simply: "During the Aeolus campaign, a new RH (relative humidity) sensor, deployed for the first time within the sonde, was found to have lag in response and did not have adequate sensitivity to vertical WV gradients. An initial view of this is provided by Fig. 14a above 5 km altitude, which will be further discussed in Sect. 4. Due to this response lag, sonde WV profiles will not be discussed in detail in this paper." Bedka et al.'s Fig. 14a seems equivalent to Fig. 6f in the current manuscript. "discussed further in Bedka et al." has been removed from the manuscript.

   I see no detailed analysis of the errors with the dropsondes nor one suggesting how to understand the limits of interpretation that can be afforded these drop sondes. Section 5.1 does attempt to explain the limits of dropsonde interpretation.

This manuscript describes no possible correction for dropsonde data. Indeed the reviewer is correct. This campaign focused on providing the best possible validation for wind lidars as it was a validation campaign for ESA/s ADM Aeolus Doppler wind lidar. The manufacturer of these sondes have identified the deficiency in the relative humidity sensor and have since then adapted the design to improve the performance of the sonde RH measurements. Although we did have discussions with the manufacturer on applying corrections to the sondes, it was beyond the scope of the funded project and the authors did not believe that corrections factors exceeding 100% would hold up in a review process. As such, we left the comparisons qualitative at best and inform the reader that more rigorous validation will be carried out in future campaigns with sondes that have traceability to community standards.

Furthermore, simply playing devil's advocate to the stated dropsonde performance, if I assumed that the dropsondes actually have no error, a plausible explanation of the differences could also be timing lag in HALO's measurements. Finally, there also appears to be a range shift of the comparisons from HALO to the satellite sensors (Figure 12). How do you sort out this discrepancy in a rigorous manner?

We thank the reviewer for examining the spaceborne sounder comparisons carefully. We reviewed the code for making these plots and found an error in the altitude arrays that was an artifact of the initial version of this comparison, which used Standard AIRS files instead of Support. This error has been corrected and the plots are updated, along with very minor tweaks to text.

**Minor Comments:**
1. Line 38: I don't see "PBL" previously defined before use here.
   Definition added.
2. Line 68: "Deutsches Zentrumvfür" should be "Deutsches Zentrum für"
   Changed.
3. Line 99: You have already defined "LaRC" in line 67.
   Fixed.
4. Line 104: As a minor follow on question to Major Question 6: Given that you need HSRL data to correct your WV absorption data for the different transmit and received spectra (and possibly your methane spectra as well?) is the WV/Methane combination truly possible? I assume this sentence is meant to say that HALO can be configured and fielded in this manner, but can you practically afford a lack of HSRL data for spectroscopic correction? Does a WV/Methane configuration alter the steps described in Figure 2?

   HALO can operate in the WV/methane configuration. As discussed above, the Doppler errors would be small. For a water vapor methane configuration we would employ the following:

   1)      For water vapor, we would carry out a Fernald retrieval to get the attenuated aerosol backscatter. Although this would not be an absolute measure of the aerosol backscatter, it does offer a correction to keep the total systematic error budget below 5%. Second, the water vapor/methane laser operates on a weaker line in the 820 nm spectral band which limits vertical coverage to below ~9km where aerosol loading is more dominant and the magnitude of the Doppler correction is lower.

   2)      Molecular Doppler correction is not required for methane IPDA as we rely on Mie Scattering only from clouds and the surface. Given we assume single scattering effects, no Doppler correction is required

   3)      When we implement the airborne version of this laser, we will fly a second laser in HALO that will give us the HSRL measurement at 532, backscatter at 1064 nm, and depolarization at both. This design is currently in the works.
5. Line 154: I don't see "ITCZ" (I assume it is Intertropical Convergence Zone) defined.

Yes, definition added.

6. Lines 191-195: There are a lot of assumptions needed for DIAL, but I think that I might include the following 2:
    1. You are assuming there are no interfering absorption species.
       This has been clarified.
    2. You are also assuming single scattering otherwise your range interval might be longer than expected.
       This has been added. However, the footprint of the DIAL channel at the surface when flying from an altitude of ~10km is 3m. So even if you had second or third order scattering, your pathlength would theoretically only increase by 3-6 meters, below our range resolution of 15 m. The optical depth accrued over this additional pathlength is within the uncertainty of our 15 m range resolution.

7. Lines 198-199: Why break out uncertainties in absorption cross section? How is this different than a systematic error?
   Absorption cross-section was named as it encompasses several potential sources of error, but we agree that it not necessarily separate from the umbrella of systematic error. Regardless, the sentence is redundant with the text around it and has been removed for concision.

8. Lines 256-266: The discussion about resolution is mostly clear. However, I can find nowhere where the pulse width is described. I assume you are oversampling, which is fine, but the pulse will serve to smooth the features you see. It also seems (though I don't know for sure) like it should also add range correlations to your data that affect your NSF. It is arguably unclear to say you have 1.25 meter timing resolution and 15 meter detector bandwidth, when some portion of your range resolution and smoothing might originate from a larger laser pulse. Additionally, this impacts the 45 meter standoff distance you need from clouds and the ground (line 298). I would suggest adding the pulse width to Table 1.
   The 935 nm, 532 and 1064 nm pulse widths are 11 ns, 18 ns, and 22 ns (full width at half maximum), resulting in unambiguous range resolution of approximately 1.65 m, 2.7 m, and 3.3 m, respectively, in air. The reviewer is correct that we oversample the pulse at the native 1.25 m vertical resolution. However, we digitally filter the 532 nm channel to 15 m resolution (the electronic bandwidth is limited to 15 m vertical resolution for the 935 and 1064 nm channels) and downsample the data from all three wavelengths to 15 m to reduce file size and ensure data from each vertical bin is uncorrelated. We do all of our subsequent calculations (including NSF) at this native 15 m resolution so range correlations are negligible.
   A sentence summarizing this has been added to the end of that paragraph, and pulse widths have been added to Table 1.

9. Figure 1: Listing the altitudes (0 and 12 km) used for your Hitran reconstruction is less helpful than the temperature and pressure in my opinion. Did I miss a reference to a standard atmosphere model where the temperature and pressure are known and linked to height?
   Figure 1a is intended to be illustrative. The temperature and pressure associated with the cross-sections were derived using a model atmosphere and have been included in the figure caption.

10. Table 1: What is the receiver field of view of the 1064 nm channel?
    1000 µrad. This has been added to Table 1.

11. Figure 2: The top gray box implies to me that you do everything highlighted in yellow for both online and offline for 3 sets of DIAL wavelength pairs. This would be 6 sets of calculations. I assume this is a misunderstanding and you really only calculate look up tables once. Is that true? If so, I suggest modifying this figure to suggest preprocessing in yellow is done for each wavelength then each wavelength is used.

That assumption is correct; the signal for each wavelength is handled independently until the DIAL calculation. Wording in the figure has been changed for clarity.

12. Figure 2: How do you apply Doppler corrections to a spliced profile? Your DIAL wavelength pairs should be differently sensitive to this effect. Should the corrections be applied to each wavelength pair before splicing?

The reviewer is correct that the Doppler correction is applied separately to each wavelength. However, the Doppler correction code used here implements splicing of the wavelength pairs in an identical manner to the WV profiles themselves, yielding a single curtain of Doppler correction factors that apply to the final product. Details and the exact order of operations of this seemed unnecessary and tedious to lay out here. A small statement to elucidate has been added to the end of the Doppler section of the text (Section 4.4) but Figure 2 has been left as-is.

13. Lines 443-447: I find this explanation a bit confusing. Are you using a different Angstrom exponent for aerosols and molecules? Presumably you are using 4 for the molecular channel and something else for the aerosol. Is that scene dependent for aerosol type? What range of values are you using?

This is only referring to the angstrom exponent for aerosol (molecular scattering can be calculated from theory). A value is calculated for each range bin from the 532 nm and 1064 nm data. That section of the text has been rewritten for clarity, and now reads: "The aerosol backscatter profile at 935 nm is estimated from the backscatter angstrom exponent at 532 nm / 1064 nm (Burton et al. 2012). This angstrom exponent comes from the HSRL 532 nm aerosol backscatter and the 1064 nm aerosol backscatter retrieved using Equation 10 of Hair et al. (2008)."

14. Figure 9: Do both of these instruments measure WV in terms of $g/m^3$? If so, they both require conversion to include the mass of air. If so, would it be more reasonable to compare as $g/m^3$?

The reviewer is correct, both instruments measure number density or absolute humidity. The choice to compare using specific humidity was in part arbitrary, however, we see value in comparing the geophysical quantity that we most often expect the end user to employ. Additionally, given that each instrument uses atmospheric state from different sources (MERRA-2 for HALO vs in-situ for DLH) to convert from absolute to specific humidity, the current analysis using specific humidity provides a more conservative estimate of uncorrelated errors between the two measurements. As such, we believe Figure 9 should be left as is.

15. Figure 9: As Figure 9 results from 3 separate measurements spliced together, it seems like it would be reasonable to break out the source of the measurement by wavelength pair. Because this comparison is so close to the plane, I would think this data is heavily dominated by the DIAL wavelength pair $\lambda_1/\lambda_2$. That said, that wavelength pair is not really what is doing the bulk of the data measurements in the $10^{-1}$ to $10^1$ range in your standard operating concept.

Wavelength pair $\lambda_1/\lambda_2$ does dominate the comparison, as you surmised. While the measurement principle and instrument design are such that unique problems are not expected from the other wavelength pairs, we will be rigorously assessing lower tropospheric data with future validation opportunities. We have added the following two sentences to the text, which already had mentioned that this comparison is mostly limited to conclusions on UT/LS WV:

"It should also be noted that due to the high-altitude nature of this comparison, 86% of the observations are with HALO wavelength pair $\lambda_1/\lambda_2$, and another 7% are in the splicing region of $\lambda_1/\lambda_2$ and $\lambda_2/\lambda_3$. When considering only the remaining 7% which are all pair $\lambda_2/\lambda_3$ and constitute most points >0.04 g/kg, bias is still very low at -0.005 g/kg."

16. Figure 9: Following on to the above comment, are you worried about the offset of 0.03 g/kg? That sounds tiny at first over all 3 orders of magnitude, but it is something like 30% bias on your driest measurements. Do you expect nearly stratospheric data measurements to increase by this quantity given the 400 or so meter difference in range?

The following sentence has been added to the text: "The low end of the comparison, $\leq 10^{-2}$ g/kg, most clearly shows a moist bias for HALO in Fig. 9, but this data is sourced entirely from 4-5 UTC on 23 April (Fig. 8b) where the aircraft was clipping the tropopause and large moisture gradients near the aircraft appear to be influencing the comparison. Removing this flight from the comparison drops the HALO wet bias by about one order of magnitude, to $6.8 \cdot 10^{-4}$ g/kg."

*Suggestions:*
1.  Line 13: Suggest changing "…uses four wavelengths at 935…" to "…near 935…"
    Changed.
2.  Lines 69-70 and 101-102: There is repetition here where you say in effect that HALO is the successor or LASE. Suggest removing one.
    Second removed.
3.  Line 146: Should "…generally fell with a 5%…" be "…within a 5%…"?
    Yes, changed.
4.  Line 167: Is NWP ever used again? If not, I would suggest removing this definition.
    Done.
5.  Line 200: I would suggest using "e.g." before your reference list here.
    Done.
6.  Line 215 (Eq. 3): Do you need some sort of reference here to scale the optical depth to account for laser power output and receiver optical path and sensitivity issues? If you take data from Figure 5 and mix up high and low sensitivity channels, you would get different answers.
    For the optical depth used in setting the splicing regions between wavelength pairs, yes, some sort of normalization could be used in principle. In practice, however, an absolute optical depth from the aircraft to range r is not required, so normalization to address differential transmit power or receiver transmission is not required. A range beyond the lidar full overlap region of ~400 m could be used to normalize the on and off signals to any differential transmission effects (accumulated OD between 0 and 400 m is negligible). Eq. 3 has been changed to OD between r and r+Δr, now including a generic normalization term.

    When multiple gains are involved, such as the surface return retrieval from Figure 5, the ratio of online to offline for each channel is still present in the DIAL equation, so differences in the detection chain or transmit power between the on and off wavelengths cancel out. For example, consider Eq. 1 where the two instances of P(λ,r+Δr) are the low gain channel and P(λ,r) is high gain.
7.  Line 217-218: Suggest removing the optimal optical depth being 1.1. Your analysis right below more accurately accounts for systematic issues and more completely describes your target OD.
    Done.
8.  Lines 242-243 and Figure 1b: Why are the count profiles shown from the low sensitivity setting? It seems like an unnecessary departure from your convention of only using high sensitivity data. I would suggest keeping all your figures constant and just using the High/High data here.
    This sounds like a misunderstanding. Figure 1b does use the High/High data, and the text is correspondingly referring to the "low end of the dynamic range" meaning the noise floor for the High/High channels, not separate detection channels. A couple words have been changed in the text to clarify.
9.  Line 408: Is "IPDA" ever used again? If not, I would suggest removing this definition.
    Done.

*Suggested references:*

- Hayman et al: "Optimization of linear signal processing in photon counting lidar using Poisson thinning," Opt. Lett. 45, 5213-5216 (2020)
- Späth et al: "Minimization of the Rayleigh-Doppler error of differential absorption lidar by frequency tuning: a simulation study," Opt. Express 28, 30324-30339 (2020)

---

## Author Response (AR2)

Response to Anonymous Referee #2 Comment on amt-2021-229 first revision
(Referee report submitted on 03 Dec 2021)

The authors graciously thank the reviewer for their thoughtful and detailed feedback, after which the manuscript has undoubtedly improved. Reviewer comments below are in green, while responses are in black.

*Summary:*
The submitted manuscript is a revised version of the original manuscript considered for publication. The details remain largely the same from version 1 to version 2. The manuscript presents an outline of the data processing methodology for the HALO instrument in the WV and HSRL configuration as well as needed hardware details, when relevant to the processing structure. The scope of the manuscript is largely similar to the original submission and is still well within the desired scope for AMT.

Overall, I believe the authors' revisions have been significant and positive. The revisions to this manuscript have alleviated a number of my original concerns relevant to this manuscript. The primary concern I had related to the quality of the ancillary data set have been well addressed by pivoting the scope to focus away from a validation. As a demonstration, the results and caveats with the available ancillary data set are exceptional. Second, the treatment of errors in this manuscript feels more complete and, while the changes are minor, they seem conclusive. The added elements cover more familiar issues related to DIAL measurements and answer a number of questions the original draft did not.

One lingering concern I have is how definitive this resource will be and its broader impact. Given that the content has not changed significantly from draft version 1 to draft version 2, the noted overlap with Bedka et al. 2021 is still prominent and much of the instrument's technical detail is still lacking. Said differently: the scope of the Bedka et al. 2021 paper forms a partial demonstration. Therefore, this demonstration of the instrument's capabilities is less significant than a single resource. This reduces the impact of this manuscript substantially in my opinion. In general, the authors' decision to publish the still to be written instrument paper after instrument demonstration, which itself was submitted after the full campaign description, should not be encouraged for the future. Assuming publication, there would now be 2 demonstrations of a single system, whose hardware is nowhere conclusively described, with reasonably similar author list using the same data set in the same journal in the same year. However, in my opinion, this is probably not reason to halt the publication of this work.

I would suggest that a few minor revisions would help clarify a couple of loose ends. Beyond that, it is my opinion that the manuscript should be published.

*Minor Comments:*
1) Page 2, Lines 52-54: This sentence currently reads "WV DIAL…only measures one species…". Suggest dropping the WV to generalize and remove redundancy.
   This change has been made.
2) Page 2, Lines 57-58: I take issue with this statement. This sentence suggests to me as the reader that adding multiple wavelength pairs is a general solution to limited DIAL dynamic range. This is of very limited practical benefit from ground based vertical profilers assuming somewhat canonical water vapor distributions. You need the rather special condition where WV is increasing with increasing range from the sensor that is really only possible from high-altitude flight. Please either alter this sentence to narrow the scope or remove.

This ability to leverage multiple wavelength pairs is an advantage of airborne DIAL over a ground-based DIAL. To clarify, the antecedent phrase has been added: "For nadir-pointing airborne DIAL, WV typically increases with range, and thus the challenges of signal attenuation and limited dynamic range with a single wavelength pair can be overcome by utilizing multiple wavelength pairs distributed along the side of the WV absorption line."

3) Page 3, Line 89: Is "n.d." "no date"? If so, should the date not be the one when the dataset's DOI was established?

The date of creation has been added (2020).

4) Page 3, Line 91: For context, it might be nice to give the number of flight hours here.

43 flight hours, this has been added.

5) Page 6, Line 182: The antecedent of "These sources of error" was, I believe, removed from version 1 to version 2. This seems unclear as it currently stands.

The antecedent phrase "These sources of error" is included in this version of the manuscript.

6) Page 7, Lines 201-203: I find it simpler to understand this sentence if you add the word "cumulative" to the first introduction to OD, i.e "The maximum cumulative WV OD …". Either way, that you have a complete profile and single range cell optical depth in the same sentence seems to me to be a point of some potential confusion.

"Cumulative" has been inserted as suggested.

7) Figure 1: A grid might help your reader out here on panels b and c. At first I was wondering why the Random error would increase near the bottom of the profile with increasing signal (due to the aerosol layer). I now see that I had mistakenly shifted the peak of the signal and peak of the error in my head to coincide in range. When I broke out a ruler as an altitude reference, that helped my understanding. A grid seems like a reasonable preventative measure here.

Grids have been added.

8) Figure 2: Do you apply the Doppler corrections at the same range spacing as the WV data? Said differently, do you use the same range resolution grid for WV and aerosol? I assume the lower resolution WV data is corrected with averaged HSRL data over the same range cell, but I don't see that stated. Did I miss that explanation?

The WV and aerosol signals and products are on the same altitude grid (with 15 m vertical spacing) as detailed around line 250.

Calculating the Doppler correction is done as the reviewer assumes, with the same range cell size used for averaging HSRL data and the DIAL calculation. This sentence has been added to Lines 467-468 to clarify: "The aerosol backscatter ratio is averaged to match the resolution of the DIAL calculation."

9) Page 15, Line 384-386: What is the maximum OD allowed for the transition from the second wavelength pair to the third? It is clear that the first transition occurs from OD = 1 to OD = 1.6. I would suggest that the second transition stopping point should be added for clarity.

The next sentence contains this information: "The second wavelength pair is then used alone until the OD range 1.0 to 1.5, wherein again a linearly weighted average controls the transition to the third wavelength pair."

10) Page 15, Line 387: This sentence might be clearer by adding that the ±0.1 refers to optical depth. It is implied but I would suggest making it explicit.

This clarification has been added.